# ON PARTIAL PROTOTYPE COLLAPSE IN CLUSTERING-BASED SELF-SUPERVISED LEARNING

## ABSTRACT

A prominent self-supervised learning paradigm is to model the representations as clusters, or more generally as a mixture model. Learning to map the data samples to compact representations and fitting the mixture model simultaneously leads to the representation collapse problem. Regularizing the distribution of data points over the clusters is the prevalent strategy to avoid this issue. While this is sufficient to prevent full representation collapse, we show that a partial prototype collapse problem still exists in methods using a prototypical formulation, that leads to significant redundancies in the prototypes. Such prototype redundancies serve as shortcuts for the method to achieve a marginal latent class distribution that matches the prescribed prior. We show that by encouraging the model to use diverse prototypes, the partial prototype collapse can be mitigated. Effective utilization of the prototypes enables the methods to learn more fine-grained clusters, encouraging more informative representations. We demonstrate that this is especially beneficial when pre-training on a long-tailed fine-grained dataset.

## 1 INTRODUCTION

Self-supervised learning (SSL) is an effective approach to learn representations from unlabelled datasets. SSL methods have progressed rapidly in recent years and even surpassed the performance achieved by supervised training on several downstream tasks (Grill et al., 2020; Chen et al., 2021; Caron et al., 2021; Zhou et al., 2022; He et al., 2022). Broadly, SSL methods can be categorized into contrastive and non-contrastive methods. Garrido et al. (2023) show that several non-contrastive methods that use covariance regularization (Zbontar et al., 2021; Bardes et al., 2022; Ermolov et al., 2021) are also contrastive in terms of the dimensions instead of the samples.

In sample-contrastive methods, all data samples repel all other data samples resulting in an approximately uniform distribution of representations in the latent space (Wang & Isola, 2020). This formulation requires large batch sizes to perform well but this limitation can be overcome by using specialized techniques like memory banks (He et al., 2020; Misra & Maaten, 2020). Recent state-of-the-art SSL methods Grill et al. (2020); Chen & He (2021); Zhou et al. (2022); He et al. (2022) use Vision Transformers (Dosovitskiy et al., 2021) and non-contrastive training methods. The prototypical formulations used in the DINO family of methods Caron et al. (2021); Li et al. (2022a); Zhou et al. (2022); Govindarajan et al. (2023); Oquab et al. (2023) enable data samples belonging to the same semantic cluster to concentrate while only repelling other clusters. Such methods are shown to learn representations that are effective at nearest neighbor tasks and few-shot learning.

A common problem in this family of methods is the representation collapse. This originates from the simultaneous learning of the image representations as well as the clustering parameters for the representations. All existing methods regularize the marginal latent class distribution in order to prevent collapse. We show that these methods are still affected by a partial prototype collapse (i.e. some groups of prototypes converge to the same vector), resulting in much fewer unique prototypes compared to the initialized number. Moreover, varying the hyperparameter for the number of prototypes has limited effect on the number of unique prototypes. The consequence is that the number of clusters learned by the method cannot be reliably controlled through the hyperparameter. Hence, it is thus far unclear what impact varying the number of clusters will have on these methods.

**Contributions:** We formally define a partial prototype collapse and demonstrate its occurrence in the DINO family of methods, the most prominent clustering-based SSL methods currently. We propose

KoLeo-proto regularization to prevent such a collapse by explicitly encouraging diverse prototypes by maximizing their differential entropy. As the name suggests, we use the Kozachenko-Leonenko differential entropy estimator (Kozachenko & Leonenko, 1987). Recently, Oquab et al. (2023) proposed to add a similar KoLeo regularization but in the context of maximizing the differential entropy of the data representations instead, which we refer to as KoLeo-data. However, this comes with the same limitation as in contrastive learning that data representations from the same semantic cluster also repel each other. Through empirical findings, we argue that it is better to maximize the diversity of the prototypes, while still allowing data representations to concentrate in clusters.

We find that effective utilization of the prototypes marginally improves performance in ImageNet downstream tasks, especially few-shot classification. However, we observe a trade-off that exists between few-shot learning performance on the pre-training dataset and transfer performance, that is consistent with other methods that also report improved ImageNet few-shot learning performance. For long-tailed fine-grained pre-training datasets such as iNaturalist-2018, we observe a clear performance improvement when classifying the same dataset without affecting the transfer performance.

## 2 BACKGROUND

SSL methods are characterized using pretext tasks. The DINO-family of methods (Caron et al., 2021; Zhou et al., 2022; Assran et al., 2022; 2023; Li et al., 2022a; Govindarajan et al., 2023) use the pretext task of assigning data to $K$ latent classes with multi-view class consistency. Object-centric datasets like ImageNet (Deng et al., 2009) are multi-view consistent and benefits from self-distillation training (Allen-Zhu & Li, 2023). Consider an encoder model that produces a $L^2$-normalized representation $\boldsymbol{y} = g_{\boldsymbol{\theta}}(\boldsymbol{x})$ such that $\|\boldsymbol{y}\| = 1$, for a data point $\boldsymbol{x}$ using parameters $\boldsymbol{\theta}$. The probability of assigning a data point to a latent class $k$ under the assumption of a latent class prior $\pi_k$ is given by:

$$P_k(\boldsymbol{y}) = \Pr(z = k|\boldsymbol{y}) = \frac{\pi_k \Pr(\boldsymbol{y}|z = k)}{\sum_{j=1}^K \pi_k \Pr(\boldsymbol{y}|z = j)}.$$

With a uniform class prior $\pi_k \equiv 1/K$ (which is true in most prior work (Assran et al., 2022)), Govindarajan et al. (2023) showed that the prototypical formulation in the DINO family corresponds to a von Mises-Fisher mixture model, with parameters $\{\boldsymbol{\mu}_k, \kappa_k\}$ and a normalization constant $C_p(\kappa_k)$

$$P_k(\boldsymbol{y}) = \frac{C_p(\kappa_k) \exp\langle\kappa_k\boldsymbol{\mu}_k, \boldsymbol{y}\rangle}{\sum_{j=1}^K C_p(\kappa_j) \exp\langle\kappa_j\boldsymbol{\mu}_j, \boldsymbol{y}\rangle}. \tag{1}$$

Here, $\boldsymbol{\mu}_k$ is the mean vector (a.k.a prototype) with $\|\boldsymbol{\mu}_k\| = 1$ and $\kappa_k > 0$ is the precision, which is a measure of concentration around the mean vector. The pre-training objective minimizes the KL-divergence between the latent class distributions of multiple views of each image. The representation learning parameters $\boldsymbol{\theta}$ as well as the mixture model parameters $\{\boldsymbol{\mu}_k, \kappa_k\}$ are learned simultaneously. This task has a trivial solution where all data points can be mapped to the same representation and be perfectly modelled using a single mixture component. To prevent this collapse, it is essential to add some form of regularization to the training objective. The regularization techniques used in such methods can be motivated using the two requirements: (i) *the model should learn distinct clusters* and (ii) *spread the data over all these clusters*. The collapse where one or a few components dominate violates requirement-(ii). The collapse of individual probability distributions to uniform distributions implies that all the prototypes are equidistant from all the data representations. In practice, such a collapse leads to all prototypes collapsing to the same vector, which violates requirement-(i).

**Connection to contrastive learning:** Contrastive learning typically uses the normalized temperature-scaled cross entropy loss based on cosine similarities. Then, the probability distribution of a query representation $\boldsymbol{y}_q$ being similar to a set of candidate representations $\boldsymbol{y}_k$ is defined as:

$$P_k(\boldsymbol{y}_q) = \frac{\exp(\langle\boldsymbol{y}_q, \boldsymbol{y}_k\rangle/\tau)}{\sum_{j=1}^K \exp(\langle\boldsymbol{y}_q, \boldsymbol{y}_j\rangle/\tau)}. \tag{2}$$

SimCLR (Chen et al., 2020) uses candidate representations from the same batch which requires large batch sizes to achieve good performance. MoCo (He et al., 2020) overcomes this limitation by using a memory bank of representations. Comparing Eq. (2) and Eq. (1), one can observe that the prototypes in DINO can be viewed as exemplary representatives of the dataset, replacing the memory bank. Based on this interpretation, the DINO family of methods can be viewed as a sparse variant of sample-contrastive methods (Garrido et al., 2023).

## 3 MARGINAL LATENT CLASS DISTRIBUTION

Before discussing a newly identified mode of collapse in the next section, we review some of the regularization techniques proposed in the literature to avoid collapse. We define the marginal latent class distribution (MLCD) as the probability vector with elements, $\bar{p}_k = \mathbb{E}_{\boldsymbol{x}}[P_k(g_{\boldsymbol{\theta}}(\boldsymbol{x}))]$. To our knowledge, all existing methods avoid representation collapse by regularizing the MLCD. Specifically, the MLCD is encouraged to match a prescribed prior distribution. A uniform prior is the default choice except for Assran et al. (2023), who propose a power law distribution to better adapt the model to long-tailed data. In a self-distillation setup, the MLCD can be encouraged to match a prior distribution either by adjusting the teacher/target distributions or by adding a penalty on the online/student distributions. In such regularization methods, sharpening is necessary to prevent individual probability distributions from collapsing to uniform distributions (Caron et al., 2021).

Adjusting the target distributions such that the MLCD matches a prior distribution can be posed as an entropy-regularized optimal transport problem, which can be efficiently solved using the Sinkhorn-Knopp (SK) algorithm (Cuturi, 2013). SK is typically run for a few iterations (typically at least 3) and adds a small but noticeable computational overhead. Caron et al. (2021) proposed a simpler and computationally efficient method to adjust the target distributions, known as centering. A key distinction between Sinkhorn-Knopp and centering is that they adjust the target distributions $P_k^{(\text{target})}(\boldsymbol{y})$ based on batch estimates and moving average estimates of the MLCD, respectively. On the other hand, Assran et al. (2022; 2023) add a prior-matching penalty on the batch-estimates of MLCD obtained from the online distributions $P_k^{(\text{online})}(\boldsymbol{y})$. The penalty is defined as the KL divergence between the MLCD and the prior distribution. With a uniform prior, this is equivalent to maximizing the entropy of MLCD, known as mean entropy maximization (ME-MAX).

**Is the centering adjustment ad-hoc?** At first glance, the centering adjustment in DINO might appear somewhat ad-hoc. However, we find that the probability centering as formulated by Govindarajan et al. (2023) is closely connected to SK. Consider a batch of $B$ logit scores over $K$ latent classes $\boldsymbol{L} \in \mathbb{R}^{B \times K}$ and corresponding probability distributions $\boldsymbol{P}$. The SK adjusted (1 iteration) probability distributions are obtained as follows (refer A.2 for derivation):

$$\tilde{\boldsymbol{P}}_{b,k}^{(\text{sk1})} = \frac{\exp(\boldsymbol{L}_{b,k} - \log(\frac{1}{B}\sum_b \boldsymbol{P}_{b,k}))}{\sum_{j=1}^{K} \exp(\boldsymbol{L}_{b,j} - \log(\frac{1}{B}\sum_b \boldsymbol{P}_{b,j}))}. \tag{3}$$

On the other hand, the probability centered distributions are obtained as follows, where the centering parameter $c_k$ is calculated as a moving average estimate with momentum parameter $m$:

$$\tilde{\boldsymbol{P}}_{b,k}^{(\text{pc})} = \frac{\exp(\boldsymbol{L}_{b,k} - c_k)}{\sum_{j=1}^{K} \exp(\boldsymbol{L}_{b,j} - c_j)}, \qquad c_k \leftarrow mc_k + (1-m)\log\left[\frac{1}{B}\sum_{b=1}^{B} \boldsymbol{P}_{b,k}\right]. \tag{4}$$

Comparing Eq. (3) and Eq. (4), we observe that probability centering is equivalent to one iteration of Sinkhorn-Knopp with the key distinction that the logit adjustment is calculated as a moving average instead of a batch estimate. We investigate this numerically in section 6.1.

## 4 PARTIAL PROTOTYPE COLLAPSE

Regularizing the MLCD enables the methods to meet the requirement of spreading data over clusters. However, since the MLCD depends on both the data representations and the prototypes, the prototypes can be manipulated in such a way that the MLCD matches the prior distribution. In other words, given a set of frozen data representations, a method can achieve MLCD matching a prior distribution simply by modifying the prototypes. Sharpening prevents the extreme case when all prototypes collapse to the same vector. However, except for this limited guardrail, the existing regularization techniques do not ensure that the methods learn unique prototypes. We define the term *partial prototype collapse*, where only a significantly small proportion of the learned prototypes are unique.

**Definition 4.1** (Partial prototype collapse). Consider the set $W = \{\boldsymbol{\mu}_k : k = 1, ..., K\}$ of $K$ prototype vectors, $\boldsymbol{\mu}_k$ such that $\|\boldsymbol{\mu}_k\| = 1$. A *partial prototype collapse* (of degree $M$ and $\epsilon$ distance) is said to have occurred if there exists a set of $M$ disjoint partitions of prototype vectors $V_m \subset W$, $m = 1, ..., M$, and $M$ representative prototype vectors $\boldsymbol{v}_m \in V_m$, such that for all $m = 1, ..., M$,

Figure 1: We reassign data to only the $M$ unique representative prototypes and compute the average proportion of data assigned to prototypes having specific redundancy factors. We find that the models tend to assign a larger proportion of data to prototypes with higher redundancy factors. This holds true for standard and vMF variants of DINO and iBOT with different backbones.

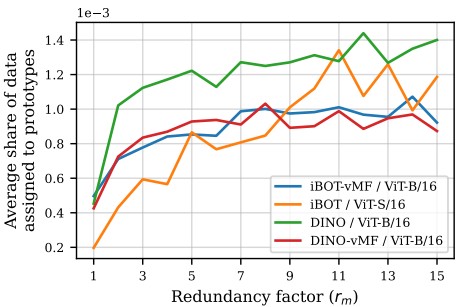

Table 1: Number of unique prototypes in existing models with $\epsilon = 0.025$ (default pre-training: ImageNet-1K, $*$: iNat-2018, $**$: ImageNet-22K)

| Backbone | Method | Initialized prototypes $(K)$ | Unique prototypes $(M)$ |
|---|---|---|---|
| ViT-S/16 | DINO | 65536 | 1078 |
| ViT-B/16 | DINO | 65536 | 804 |
| ViT-S/16 | DINO-vMF | 65536 | 1157 |
| ViT-B/16 | DINO-vMF | 65536 | 939 |
| ViT-S/16 | iBOT | 8192 | 3242 |
| ViT-B/16 | iBOT | 8192 | 875 |
| ViT-B/16 | iBOT-vMF | 8192 | 1170 |
| ViT-L/16 | iBOT | 8192 | 969 |
| ViT-B/16 | iBOT** | 8192 | 1241 |
| ViT-L/16 | iBOT** | 8192 | 1037 |
| ViT-S/16 | MSN* | 8142 | 3363 |
| ViT-S/16 | PMSN* | 8142 | 3005 |

$1 - \boldsymbol{v}_m^\mathsf{T} \boldsymbol{\mu}_j < \epsilon$, for all $\boldsymbol{\mu}_j \in V_m$. The set of $M$ *unique prototypes* is defined as $U = \{\boldsymbol{v}_m\}_{m=1}^M$. For each representative prototype, the *redundancy factor* $r_m$ is defined as the size of the corresponding set partition, $r_m = |V_m|$.

**Investigating learned MLCD and prototypes**: When training the prototypes and the representations simultaneously with MLCD regularization, the methods are prone to partial prototype collapse since it enables the method to spread probability mass associated with each unique prototype across its $\epsilon$-set of redundant prototypes. This acts as a shortcut to match the MLCD to the specified prior distribution. Govindarajan et al. (2023) make an empirical observation that the DINO models used significantly smaller number of unique prototypes compared to the hyperparameter $K$. However, this problem is neither studied further nor addressed by their proposed method. Based on our definition of partial prototype collapse and using a cosine distance metric, we investigate the prototypes learned by several self-supervised clustering methods that use a prototypical formulation, from SwAV (Caron et al., 2020) to iBOT (Zhou et al., 2022). In Table 1 and Table 9, we show that such a collapse exists in all the considered methods. We observe that prototypes with a higher redundancy factor tend to be assigned a larger proportion of the data samples (see Figure 1). Hence, the partial prototype collapse serves as a shortcut to achieve a MLCD closer to a uniform distribution. In addition, this means that the hyperparameter $K$ does not play its intended role of controlling the number of clusters.

## 4.1 Regularizing prototype distribution

The number of latent classes is an important choice in clustering and mixture models. In a SSL method, this choice controls the fine-grainedness of the clusters. Firstly, this controls the difficulty of the self-supervision task. Secondly, more informative representations are required to discriminate between more fine-grained latent classes. With this motivation, we believe that the number of prototypes is an important design choice in SSL as well. However, prior works have found inconsistent results when ablating for this choice, likely because of the occurrence of partial prototype collapse.

Given that we want the prototypes to be as diverse as possible, a meaningful choice is to encourage the prototypes $\boldsymbol{W} = \{\boldsymbol{\mu}\}_{k=1}^K$ to be uniformly distributed in the latent space. We propose to achieve this by maximizing the differential entropy of the prototype vectors, obtained using the Kozachenko-Leonenko estimator (Kozachenko & Leonenko, 1987; Beirlant et al., 1997; Sablayrolles et al., 2019),

$$\mathcal{L}_{\text{KP}} = h_{\text{kl}}(\boldsymbol{W}) = -\frac{1}{K} \sum_{k=1}^K \log(d_k); \quad d_k = \min_{i \neq k} \|\boldsymbol{\mu}_k - \boldsymbol{\mu}_i\|. \tag{5}$$

We efficiently compute an estimate of $h_{\text{kl}}(\boldsymbol{W})$ by randomly partitioning the prototypes into batches, that adds negligible computational overhead in terms of memory and time (see A.3 for details).

## 5    RELATED WORK

**Connection to DINOv2**: Our proposed KoLeo-proto regularization is similar to Sablayrolles et al. (2019). Recently, DINOv2 (Oquab et al., 2023) proposed the KoLeo-data regularization which uses a similar formulation but applied to spread the data representations instead of the prototypes. Hence, DINOv2 can be viewed as an interpolation between the uniformly distributed representations of contrastive learning and clustered representations of the DINO family. In contrast, KoLeo-proto preserves the clustered representations of DINO and encourages the method to learn diverse clusters.

**Regularizations in clustering-based SSL**: We provide an extended discussion of clustering-based SSL methods in A.1 and focus our discussion on the regularization methods in this section. A common limitation of simultaneously learning representations and clustering them is that there are degenerate solutions that perfectly solve the clustering task but fail to learn informative representations. Caron et al. (2018) proposed uniform pseudo-label sampling that is equivalent to weighting the loss contribution of an input by the inverse of its assigned cluster's size. Asano et al. (2020); Caron et al. (2020) viewed the clustering task with MLCD regularization as an entropy regularized optimal transport problem and use the Sinkhorn-Knopp algorithm to assign pseudo-labels to data points (Cuturi, 2013). This is shown by Assran et al. (2023) to encourage the MLCD to match a uniform prior. While Sinkhorn-Knopp requires multiple iterations for convergence, a simpler and computationally cheaper approach known as centering is proposed in DINO (Caron et al., 2021) and also used in EsViT (Li et al., 2022a) and iBOT (Zhou et al., 2022). Govindarajan et al. (2023) proposed probability centering, that computed the centering parameter in the probability space instead of the logit space. Assran et al. (2023) proposed to add an explicit prior matching penalty to encourage the MLCD to align with a prescribed prior distribution. With a uniform prior we obtain mean entropy maximization (Assran et al., 2022). Methods using the prior-matching penalty and Sinkhorn-Knopp depend on batch estimates of the MLCD. On the other hand, centering uses moving average estimates; we showed the connection of probability centering to Sinkhorn-Knopp in section 3 and investigate how methods using batch and moving average MLCD estimates compare at different batch sizes in section 6.1. While all the above methods regularize the MLCD, we show the occurrence of a partial prototype collapse by investigating the prototypes learned by existing pre-trained models. We propose a new KoLeo-proto regularization to prevent this collapse and effectively utilize the prototypes.

**Pre-training on long-tail datasets**: Most SSL methods are evaluated by pre-training on ImageNet, a well-curated dataset with a uniform class distribution. To the contrary, real-world data collection often results in long-tailed distributions over visual concepts and pre-training on such datasets is of practical interest. We note that there is limited research on pre-training SSL methods on such long-tailed datasets. Caron et al. (2019) investigated pre-training on a large uncurated dataset. Recently, Kukleva et al. (2023) explored the benefits of using temperature schedules in the context of contrastive learning. Assran et al. (2023) showed that pre-training on a long-tailed dataset can benefit from choosing an appropriate long-tail prior. We investigate the impact of our proposed KoLeo-proto regularization by pre-training on the long-tailed and fine-grained iNaturalist-18 dataset in section 7. Yang et al. (2022) overcame a minority collapse issue (Fang et al., 2021) in supervised long-tailed classification with a fixed classification layer based on ETF geometry. However, this comes with the implicit assumption that all class prototypes should be equidistant which is a strong assumption for the latent classes learned in SSL. The KoLeo-proto regularization is a soft penalty that encourages the prototypes to remain distinct while still allowing semantically similar clusters to have more similar prototypes. This enables the SSL method to learn better semantically meaningful representations.

## 6    IMAGENET EXPERIMENTS

To study the MLCD and prototype regularizations, we focus on iBOT, which is a strong recent baseline among the DINO family of methods and also used as the foundation for DINOv2 (Oquab et al., 2023). We pre-train the models on the ImageNet-1K dataset (Deng et al., 2009) by modifying the public codebase of iBOT. We use the same hyperparameter settings as in iBOT for different ViT backbones (refer A.4.1 for details) and use the vMF normalized variants (Govindarajan et al., 2023), which are shown to produce stable trainings and improved performance. In A.5.3, we additionally study the DINO method with a Resnet backbone. We start with ablation experiments to choose the MLCD regularization technique in section 6.1, evaluate the impact of adding our proposed prototype regularization in section 6.2 and finally perform full-scale pre-training experiments.

## 6.1 MLCD REGULARIZATION

Firstly, we run ablation experiments to select the method to regularize MLCD. We pre-train ViT-Small/16 backbone with different MLCD regularization techniques - Sinkhorn-Knopp (SK), probability centering (PC) and mean entropy maximization (ME-MAX). For PC, we use the vMF normalized version of iBOT. For SK and ME-MAX, we chose to use a smaller teacher temperature based on a hyperparameter search (refer A.4.1 for details). We also consider three different compute budgets (2, 4 and 8 GPUs for 2 days), which allows us to evaluate the impact of batch size on these techniques. With more GPUs, we can accommodate a larger batch size. The number of epochs is adjusted such that the total number of iterations are the same for all the compute budgets. We do this to avoid the expensive process of optimizing the learning rates for each compute budget and regularization method. Overall, from Figure 2, we find that probability centering performs better than the other alternatives at different compute budgets. Interestingly, PC achieves performance on par or better than the alternatives, even at half of the compute budget (e.g. PC/4GPUs vs ME-MAX/8GPUs).

The methods discussed above have all been proposed in the literature as ways to regularize the MLCD. We argue that the main difference between them is whether the regularization is done over a single batch (SK, ME-MAX) or based on moving average statistics (PC). We observe that PC performs significantly better than the alternatives at the lowest compute budget, which uses a small batch size. As we increase the compute budget and thereby also the batch size, the gap is reduced. This indicates that PC is more robust to the choice of batch size. We conjecture that this is due to too noisy estimates of the MLCD when computed over a batch, which is not surprising considering that we estimate probability vectors in a high-dimensional space. In the following experiments, we use the vMF normalized iBOT with MLCD regularized using probability centering.

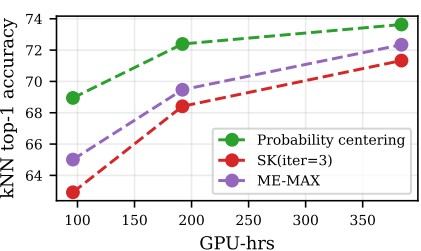

Figure 2: ImageNet top-1 kNN accuracy with different MLCD regularization approaches. Probability centering performs better than SK and ME-MAX at different compute budgets.

## 6.2 PROTOTYPE REGULARIZATION

We add our proposed KoLeo-proto regularization to the iBOT-vMF baseline, resulting in the overall loss objective, $\mathcal{L} = \mathcal{L}_{iBOT} + \lambda\mathcal{L}_{KP}$. These results are indicated by "(kp)". Similarly, we indicate the KoLeo-data regularization used by Oquab et al. (2023) as "(kd)". We use $\lambda = 0.1$, similar to DINOv2 and observe that such a small $\lambda$ is sufficient to mitigate partial prototype collapse and ensure that almost all of the initialized prototypes remain unique. In Figure 3, we compare the number of unique prototypes when we vary the initialized number of prototypes hyperparameter $K$. With the baseline and KoLeo-data regularization, changing the number of prototypes has no impact on the number of unique prototypes learned by the method, which is significantly smaller than the initialized number of prototypes. This indicates the occurrence of partial prototype collapse.

We observe that the baseline shows similar performance at different numbers of initialized prototypes. On the other hand, with KoLeo-data, the performance is worse than the baseline but continues to improve as the number of prototypes are increased. KoLeo-data encourages the data to spread on the hypersphere. Hence, data is assigned to more diverse prototypes compared to the baseline in the initial training phase. We conjecture that this initial training dynamic benefits from having more prototypes, even if many of these prototypes eventually collapse to the same vector. The performance with KoLeo-data is expected to improve further as the number of prototypes are increased, as noted by Oquab et al. (2023) when using 128K prototypes. Applying KoLeo-proto leads to improved performance over the baseline and KoLeo-data and continues to improve further as we increase the number of prototypes. We limit the maximum number of prototypes to 10240 due to computational limitations. Computing probability distributions for all the tokens over more dimensions adds a large computational overhead. However, the KoLeo regularization itself only adds a negligible computational overhead (both memory and time, cf. A.3). We observe around 0.1% improvement in accuracy when adding every 2K additional prototypes. Overall, increasing the number of prototypes from 2K to 10K results in a 0.4% improvement. Further scaling of the number of prototypes can bring larger performance gains which should be feasible with the efficient implementation in DINOv2.

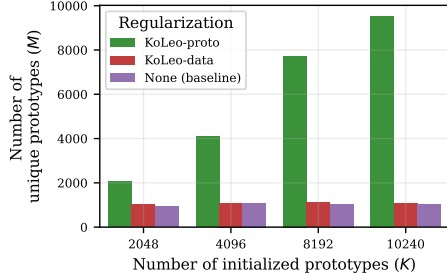 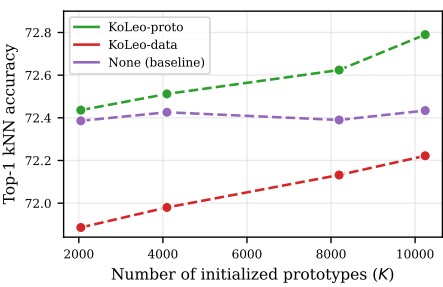

Figure 3: (left) The number of unique prototypes are similar for the baseline and KoLeo-data regularization at different number of initialized prototypes. With KoLeo-proto, most of the initialized prototypes remain unique. (right) The number of initialized prototypes has no impact on the baseline performance. With any form of KoLeo-regularization, more prototypes lead to better performance. KoLeo-proto generally produces the best performance at varying numbers of prototypes.

## 6.3 IMAGENET CLASSIFICATION

We pre-train iBOT-vMF with the KoLeo regularization applied to the prototypes for ViT-Small/16 and ViT-Base/16 backbones. To ensure fair comparison, we set the number of prototypes to 8192, similar to iBOT. Hence, any changes in performance can be associated to only our proposed KoLeo-proto regularization. In Table 2, we report the top-1 accuracies obtained using kNN and linear classification based on frozen backbone features, few-shot accuracies averaged over 3 different splits and the accuracy obtained after fine-tuning. For kNN and linear evaluation, we follow the same protocol as in DINO and iBOT. We perform few-shot evaluation similar to Assran et al. (2022) and use the provided data splits. We use the fine-tuning recipe from iBOT (Zhou et al., 2022) and BeIT (Bao et al., 2022), using a layer-wise learning rate decay. We compare against the iBOT-vMF baseline, MSN and the best performing models from WE-SSL (Ruan et al., 2023). We observe on par or marginal improvements for kNN, linear and fine-tuned classification performance. The kNN performance improvement with respect to the baseline at 8192 prototypes after full-scale pre-training mirrors the improvement (+0.2%) observed after the small-scale ablation pre-training in Figure 3. This suggests that by using an even larger number of prototypes one can improve the performance further with KoLeo-proto regularization (cf Figure 3), which we found to not be the case for the baseline in ablation experiments (unfortunately we lack the computational resources to verify this empirically).

We find larger gains for few-shot learning performance when adding KoLeo-proto to the baseline, even at 8192 prototypes. Note that the prediction head architecture and other hyperparameters are tuned in WE-SSL to achieve the best few-shot learning performance with ViT-Small/16. We do not perform any such specific tuning and this explains the significantly better results achieved by WE-SSL with ViT-Small/16. With ViT-Base/16, iBOT-vMF (kp) outperforms WE-SSL at 1% and 5 img/cls settings. Note that iBOT-vMF can be tuned similar to WE-SSL but we have not investigated this. Instead, we focus on studying the impact of effective utilization of the prototypes on general downstream performance and do not perform any tuning for a specific task.

## 6.4 TRANSFER LEARNING WITH IMAGENET PRE-TRAINING

We conduct linear classification experiments on the standard suite of datasets trained using features extracted from a frozen pre-trained model. In Table 3, we report the linear classification accuracy on the validation/test set depending on availability. We evaluate the impact of adding the KoLeo-proto regularization to the iBOT-vMF baseline. We also report results from DINO-vMF (Caron et al., 2021; Govindarajan et al., 2023), MSN (Assran et al., 2022) and WE-SSL (Ruan et al., 2023) methods. We observe on par or decreased transfer performance when adding our proposed KoLeo-proto regularization compared to the iBOT-vMF baseline. Interestingly, we note that the transfer performance decreases also in other methods that improve few-shot learning performance such as MSN (Assran et al., 2022) and WE-SSL (Ruan et al., 2023) compared to their DINO baseline (note that transfer results for ViT-Base/16 are not reported for WE-SSL). This indicates that tuning for few-shot learning performance can potentially harm transfer performance. Compared to these

Table 2: ImageNet classification with full data (kNN, linear, finetuning) and few-shot scenarios.

| Method | kNN | Linear | Finetuning | 1% data | 5 img/cls | 2 img/cls | 1 img/cls |
|---|---|---|---|---|---|---|---|
| *ViT-Base/16* | | | | | | | |
| MSN | 73.3 | 74.8 | – | 69.1 | 65.5 | 58.9 | 49.8 |
| WE-SSL | 77.2 | 78.9 | – | 71.5 | 68.3 | **62.4** | **53.7** |
| iBOT-vMF | 78.7 | 80.3 | **84.1** | 72.3 | 68.3 | 61.1 | 51.6 |
| iBOT-vMF (kp) | **78.8** | **80.5** | **84.1** | **72.7** | **69.1** | 62.0 | 52.5 |
| *ViT-Small/16* | | | | | | | |
| MSN | 74.9 | 76.6 | – | 67.2 | 62.8 | 55.8 | 47.1 |
| WE-SSL | 75.2 | 77.4 | – | **68.7** | **65.1** | **58.9** | **50.1** |
| iBOT-vMF | 75.3 | 77.9 | **82.3** | 66.4 | 60.6 | 51.1 | 40.7 |
| iBOT-vMF (kp) | **75.5** | 77.9 | **82.3** | 67.0 | 61.1 | 51.7 | 41.6 |

Table 3: Linear classification accuracy when transferred to other datasets

| Method | Cal101 | C10 | C100 | DTD | Flwrs. | Food | Pets | SUN | Avg. |
|---|---|---|---|---|---|---|---|---|---|
| *ViT-Base/16* | | | | | | | | | |
| DINO-vMF | 94.5 | 97.1 | 86.3 | **74.8** | **95.7** | 82.5 | **94.6** | 68.7 | 86.8 |
| iBOT-vMF | **95.5** | **98.0** | **88.0** | 74.7 | 94.8 | **83.6** | 93.9 | **70.2** | **87.3** |
| iBOT-vMF (kp) | 94.6 | 96.5 | 84.1 | 74.3 | 95.6 | **83.6** | 94.0 | 69.7 | 86.6 |
| MSN | 92.8 | 96.9 | 85.3 | 73.7 | 92.8 | 80.0 | 93.9 | 66.8 | 85.3 |
| *ViT-Small/16* | | | | | | | | | |
| DINO-vMF | 93.7 | 96.0 | 83.9 | 74.1 | **95.0** | 80.1 | 93.9 | 66.6 | 85.4 |
| iBOT-vMF | 94.1 | **96.7** | **84.6** | 72.8 | 94.3 | 80.3 | **94.1** | 67.3 | 85.5 |
| iBOT-vMF (kp) | 94.5 | **96.7** | 83.9 | 73.7 | 94.4 | **80.5** | 93.7 | **67.5** | **85.6** |
| MSN | 93.1 | 95.9 | 82.9 | 72.0 | 93.3 | 77.8 | 92.8 | 65.5 | 84.1 |
| WE-SSL | **94.6** | 93.8 | 81.4 | **74.9** | 93.9 | 79.1 | 92.8 | 66.5 | 84.6 |

methods, our proposed regularization leads to better transfer performance. There appears to be a trade-off between few-shot learning performance on the pre-training dataset and transfer learning performance. Currently, it is unclear why such a trade-off exists and this requires further investigation.

## 7 iNaturalist-2018 experiments

Most SSL methods are pre-trained on ImageNet which is well-curated and contains uniformly distributed data across its classes. For practical use-cases, SSL methods need to be trained on data collected in the wild, which is often long-tailed. Hence, it is of interest to study the effect of pre-training methods on long-tailed datasets as well, which has gained limited attention. We consider the iNaturalist-2018 (iNat18) dataset [1] which is around 1/3rd of the size of ImageNet and contains a long-tail distribution of data from 8142 classes. We pre-train all the models for 300 epochs using the default publicly available hyperparameters. For MSN and PMSN (Assran et al., 2022; 2023), we choose the regularization strength $\lambda$ based on a hyperparameter search (see A.4.2 for details). In Table 4, we report the top-1 classification accuracy obtained using a linear and a fine-tuned classifier. For linear classification, we follow a similar protocol as in the

Table 4: iNat-2018 classification accuracies with full data (linear probing and fine-tuning)

| Method | $M$ | Linear | Fine-tuned |
|---|---|---|---|
| *ViT-Small/16* | | | |
| DINO-vMF | 1380 | 49.7 | 68.5 |
| iBOT-vMF | 1804 | 50.1 | **69.4** |
| iBOT-vMF (kd) | 1843 | 50.5 | 69.1 |
| iBOT-vMF (kp) | 7895 | **51.1** | 69.3 |
| MSN ($\lambda = 1$) | 3363 | 43.8 | 63.5 |
| PMSN ($\lambda = 5$) | 3005 | 41.8 | 64.2 |
| *ViT-Base/16* | | | |
| iBOT-vMF (kd) | 1634 | 50.4 | 73.3 |
| iBOT-vMF (kp) | 7573 | **51.4** | **74.0** |

---

[1]This dataset was used for academic purposes only.

Table 5: Linear classification accuracy when transferred to other datasets (pre-trained on iNat-2018)

| Method | Cal101 | C10 | C100 | DTD | Flwrs. | Food | Pets | SUN | Avg. |
|---|---|---|---|---|---|---|---|---|---|
| *ViT-Small/16* | | | | | | | | | |
| iBOT-vMF | **80.3** | **87.0** | 69.2 | 66.3 | 93.6 | 66.4 | 64.2 | 47.4 | 71.8 |
| iBOT-vMF (kd) | 79.3 | 86.3 | 68.1 | 64.7 | **94.2** | 66.4 | 65.4 | 47.6 | 71.5 |
| iBOT-vMF (kp) | 80.1 | 86.7 | **69.9** | **66.8** | 93.2 | 66.6 | **65.5** | 47.4 | **72.0** |
| *ViT-Base/16* | | | | | | | | | |
| iBOT-vMF (kd) | 82.4 | 87.9 | 70.8 | 66.3 | 94.6 | 68.6 | 66.9 | 48.5 | 73.2 |
| iBOT-vMF (kp) | 82.0 | 88.7 | 72.1 | 66.3 | 94.7 | 68.7 | 68.1 | 48.7 | 73.7 |

ImageNet experiments. For fine-tuning, we use longer trainings with a smaller learning rate as in DINO (Caron et al., 2021) (see details in A.4.5). We consider iBOT-vMF as our baseline method, which significantly outperforms MSN and PMSN.

For ViT-Small model, we find that both KoLeo regularization methods bring performance benefits compared to the baseline. After evaluating the two forms of KoLeo regularization on the ViT-Base model as well, we conclude that KoLeo-proto regularization performs best. With partial prototype collapse, models learn more coarse-grained latent classes where the number of unique prototypes are less than the number of classes (see $M$ in Table 4). Then, the learned clusters are likely to have merged several of the fine-grained classes. This is mitigated when the prototypes are effectively utilized, leading to more diverse clusters and hence, more informative representations which are beneficial for long-tailed and fine-grained classification tasks. Hence, iNat-2018 pre-training benefits more from utilizing more unique prototypes compared to the Imagenet experiments. We conduct transfer learning experiments using the same evaluation protocol as with ImageNet pre-training. The results are reported in Table 5. Here, KoLeo-proto regularization performs slightly better than KoLeo-data while remaining on par with the iBOT-vMF baseline. This is in contrast to the ImageNet experiments where KoLeo-proto regularization negatively impacted transfer performance.

## 8    CONCLUSION

We identified the occurrence of a previously unnoticed mode of collapse in the DINO family of methods, termed as *partial prototype collapse* that results in significant redundancies in the prototypes. As a consequence, the hyperparameter controlling the number of prototypes did not perform its intended role of controlling the number of clusters learned by the model. We proposed the KoLeo-proto regularization to encourage the model to learn diverse prototypes. By adding our proposed regularization, we showed that most of the initialized prototypes are effectively utilized. With effective prototype utilization, scaling the number of prototypes is useful in learning better image representations of the underlying dataset. Using the same moderate number of 8K prototypes as before, we showed that few-shot learning performance can be improved and full data trainings can be marginally improved. As indicated in our ablation experiments, it seems possible that further scaling the number of prototypes can result in more significant improvements. However, we observed a worse transfer performance when learning fine-grained clusters of the pre-training dataset. This trade-off is consistent with other methods that specifically improve few-shot learning performance. Investigating this trade-off is an interesting direction for future research. On the other hand, we found that learning fine-grained clusters on a long-tailed fine-grained dataset such as iNat-2018 is more beneficial, indicated by the larger performance gains achieved using a similar number of prototypes.

We have shown that the hyperparameter for the number of prototypes can be reliably controlled using our regularization. This has broad implications on applying methods from the DINO family. One can better understand the impact of using different numbers of clusters in the self-supervised pretext task for their own dataset and method of choice. This could vary depending on the domain of the dataset and how fine-grained the semantic concepts are in that domain. Computing probability distributions over a large number of latent classes comes at a significant computational cost (see A.5.4). If indeed a small number of clusters are sufficient for some dataset, effectively utilizing fewer prototypes can help in reducing computational expenses.

## REPRODUCIBILITY STATEMENT

We use the same hyperparameter setup as iBOT (Zhou et al., 2022) which is already publicly available. Nevertheless, we provide the complete hyperparameter configurations in A.4.1 to help with reproducing our experiments correctly. We provide details about the implementation of our proposed regularization method in A.3. For other downstream tasks, we follow standard evaluation protocol of other works and differences, if any, are discussed in the experiment. We provide additional experimental details A.4.

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

# A APPENDIX

## A.1 EXTENDED RELATED WORK

**Clustering-based self-supervised learning**: Self-supervised learning based on the clustering pretext task is a promising paradigm that has proven to be successful and grown tremendously in recent years. Initial works (Caron et al., 2018; 2019; Asano et al., 2020) used a two-stage process of assigning pseudo-labels by clustering the representations and then training the representations using the pseudo-labels as targets. Caron et al. (2020) used online assignment of pseudo-labels in every batch by clustering the representations over a small window of batches. By adapting this to ViTs, Caron et al. (2021) proposed a self-distillation framework where a teacher network produced the target latent classes. Govindarajan et al. (2023) demonstrated that this objective corresponds to learning a von Mises-Fisher mixture distribution. Li et al. (2022a) extended the DINO objective to patch tokens while also leveraging efficient architectures like Swin (Liu et al., 2021). iBOT (Zhou et al., 2022) is a recent state-of-the-art method that poses the masked image modeling (MIM) task of BeIT (Bao et al., 2022) as a clustering task. Another branch of works have focused on improving the few-shot learning performance of these methods (Assran et al., 2022; Ruan et al., 2023). Recently, DINOv2 (Oquab et al., 2023) built upon iBOT by making several modifications. By pre-training on the large LVD142M dataset, DINOv2 demonstrated performance surpassing many state-of-the-art visual benchmarks at image and pixel levels.

## A.2 SINKHORN-KNOPP AND PROBABILITY CENTERING

Let the batch of $B$ logit scores be denoted as $\boldsymbol{L} \in \mathbb{R}^{B \times K}$ with corresponding probability distributions $\boldsymbol{P}$. Then, the Sinkhorn-Knopp adjusted probability distributions $\tilde{\boldsymbol{P}}$ are obtained by alternating between normalizing the rows and columns of the matrix $\exp(\boldsymbol{L})$, so that they sum up to 1. Note that the exponent function is applied element-wise to the matrix. Let the elements of the matrix be denoted as $\boldsymbol{L}_{b,k}$. Then, normalizing along the rows yields,

$$\tilde{\boldsymbol{P}}_{b,k} \leftarrow \frac{\exp(\boldsymbol{L}_{b,k})}{\sum_b \exp(\boldsymbol{L}_{b,k})} = \frac{\frac{1}{B}\exp(\boldsymbol{L}_{b,k})}{\frac{1}{B}\sum_b \exp(\boldsymbol{L}_{b,k})} = \frac{1}{B}\exp(\boldsymbol{L}_{b,k} - \log(\frac{1}{B}\sum_b \exp(\boldsymbol{L}_{b,k}))).$$

Next, normalizing $\tilde{\boldsymbol{P}}$ along the columns we obtain,

$$\tilde{\boldsymbol{P}}_{b,k} \leftarrow \frac{\exp(\boldsymbol{L}_{b,k} - \log(\frac{1}{B}\sum_b \exp(\boldsymbol{L}_{b,k})))}{\sum_{j=1}^K \exp(\boldsymbol{L}_{b,j} - \log(\frac{1}{B}\sum_b \exp(\boldsymbol{L}_{b,j})))}.$$

If we consider the initial logit scores $\boldsymbol{L}_b$ to be already normalized over the components $K$ such that $\sum_k \exp(\boldsymbol{L}_{b,k}) = 1$, then the exponents within the inner sum can be replaced with probabilities. Thus, we obtain the probability distributions after 1 iteration of Sinkhorn-Knopp adjustment as,

$$\tilde{\boldsymbol{P}}_{b,k}^{(\mathrm{sk1})} \leftarrow \frac{\exp(\boldsymbol{L}_{b,k} - \log(\frac{1}{B}\sum_b \boldsymbol{P}_{b,k}))}{\sum_{j=1}^K \exp(\boldsymbol{L}_{b,j} - \log(\frac{1}{B}\sum_b \boldsymbol{P}_{b,j}))}.$$

On the other hand, the probability centered distributions proposed by Govindarajan et al. (2023) are obtained as follows, where the centering parameter $c_k$ is calculated as a moving average estimate with momentum parameter $m$:

$$\tilde{\boldsymbol{P}}_{b,k}^{(\mathrm{pc})} = \frac{\exp(\boldsymbol{L}_{b,k} - c_k)}{\sum_{j=1}^K \exp(\boldsymbol{L}_{b,j} - c_j)}, \qquad c_k \leftarrow mc_k + (1-m)\log\left[\frac{1}{B}\sum_{b=1}^B \boldsymbol{P}_{b,k}\right].$$

Comparing the above expressions for $\tilde{\boldsymbol{P}}_{b,k}^{(\mathrm{sk1})}$ and $\tilde{\boldsymbol{P}}_{b,k}^{(\mathrm{pc})}$ (Eq. (3) and Eq. (4) in section 3), we observe that probability centering is equivalent to one iteration of Sinkhorn-Knopp with the key distinction that the logit adjustment is calculated as a moving average instead of a batch estimate.

## A.3 KOLEO PROTOTYPES IMPLEMENTATION

Given a set of $K$ prototypes $\boldsymbol{W} \in \mathbb{R}^{K \times D}$, to compute the KoLeo estimate of the differential entropy of the prototypes $h_{\mathrm{kl}}(\boldsymbol{W})$, we require computing nearest neighbor distances for each of

the prototypes. This can be memory intensive when a large number of prototypes are used. Note that this is not a problem in the case of DINOv2 (Oquab et al., 2023), as the KoLeo objective is computed between the $B$ data representations in the batch ($B$ is typically much smaller than $K$). Instead, we resort to a stochastic estimate when calculating the loss objective in each batch. For each batch, we randomly partition the prototypes into disjoint partitions containing 2048 prototypes each, $\boldsymbol{W} = \{\boldsymbol{W}_1, ..., \boldsymbol{W}_T\}, \boldsymbol{W}_t \in \mathbb{R}^{2048 \times D}$. Then, we compute the KoLeo estimate as follows: $h_{\mathrm{kl}}(\boldsymbol{W}) = \sum_{t=1}^{T} h_{\mathrm{kl}}(\boldsymbol{W}_t)$. This efficient batched implementation adds negligible computational overhead, in terms of both memory and time (15 MB additional GPU memory when $K = 8192$ and unchanged image throughput).

## A.4 EXPERIMENTAL DETAILS

### A.4.1 HYPERPARAMETER SETTINGS

The complete hyperparameter configuration for full-scale iBOT-vMF pre-trainings on ImageNet using ViT-Small/16 and ViT-Base/16 models are provided in Table 6. For pre-training on iNaturalist-2018, we use a similar hyperparameter configuration except that we use pre-train both ViT-Small/16 and ViT-Base/16 models for 300 epochs. The complete hyperparameter configurations for MSN and PMSN pre-trainings on the iNaturalist-2018 dataset using the ViT-Small/16 model are provided in Table 7.

Table 6: Hyperparameter settings for iBOT

| Hyper-parameter | ViT-Small/16 | ViT-Base/16 |
|---|---|---|
| training epochs | 800 | 400 |
| batch size | 1024 | 512 |
| learning rate | 2e−3 | 1.5e−3 |
| warmup epochs | 10 | 10 |
| freeze last layer epochs | 1 | 3 |
| min. learning rate | 1e−6 | 2e−6 |
| weight decay | $0.04 \to 0.4$ | $0.04 \to 0.4$ |
| stochastic depth | 0.1 | 0.1 |
| gradient clip | 3.0 | 0.3 |
| optimizer | adamw | adamw |
| shared head | ✓ | ✓ |
| fp16 | ✓ | ✓ |
| momentum | $0.996 \to 1.0$ | $0.996 \to 1.0$ |
| global crops | 2 | 2 |
| global crops scale | $[0.25, 1.0]$ | $[0.32, 1.0]$ |
| local crops | 10 | 10 |
| local crops scale | $[0.05, 0.25]$ | $[0.05, 0.32]$ |
| head mlp layers | 3 | 3 |
| head hidden dim. | 2048 | 2048 |
| head bottleneck dim. | 256 | 256 |
| norm last layer | ✗ | ✗ |
| num. prototypes | 8192 | 8192 |
| vmf normalization | ✓ | ✓ |
| centering | probability | probability |
| koleo reg. strength | 0.1 | 0.1 |
| teacher temp. | $0.04 \to 0.07$ | $0.04 \to 0.07$ |
| temp. warmup epochs | 30 | 50 |
| student temp. | 0.1 | 0.1 |
| pred. ratio | $[0.0, 0.3]$ | $[0.0, 0.3]$ |
| pred. ratio variance | $[0.0, 0.2]$ | $[0.0, 0.2]$ |
| pred. shape | block | block |

Table 7: Hyperparameter settings for MSN / PMSN

| Hyper-parameter | MSN | PMSN |
|---|---|---|
| training epochs | 300 | 300 |
| batch size | 1536 | 1536 |
| learning rate | 6e−3 | 6e−3 |
| warmup epochs | 15 | 15 |
| min. learning rate | 1e−6 | 2e−6 |
| weight decay | $0.04 \rightarrow 0.4$ | $0.04 \rightarrow 0.4$ |
| stochastic depth | 0.1 | 0.1 |
| gradient clip | 3.0 | 3.0 |
| optimizer | adamw | adamw |
| fp16 | ✗ | ✗ |
| momentum | $0.996 \rightarrow 1.0$ | $0.996 \rightarrow 1.0$ |
| random crops | 1 | 1 |
| local crops | 10 | 10 |
| patch drop rate | 0.15 | 0.15 |
| head mlp layers | 3 | 3 |
| head hidden dim. | 2048 | 2048 |
| head bottleneck dim. | 256 | 256 |
| norm last layer | ✓ | ✓ |
| num. prototypes | 8142 | 8142 |
| kl penalty weight ($\lambda$) | 1.0 | 5.0 |
| teacher temp. | 0.025 | 0.025 |
| sinkhorn teacher | ✓ | ✓ |
| temp. warmup epochs | 30 | 50 |
| student temp. | 0.1 | 0.1 |

### A.4.2 MSN AND PMSN DISCUSION

When pre-training on the iNaturalist-2018 dataset using the ViT-Small/16 model, we run hyperpa-rameter sweeps to select suitable values for the KL penalty strength parameter $\lambda$. We consider the values $\{1.0, 5.0, 15.0\}$. Based on the linear probing results shown in Table 8, we select $\lambda = 1.0$ for MSN and $\lambda = 5.0$ for PMSN. Using a higher $\lambda$ with MSN strongly encourages the MLCD to match a uniform prior distribution. When the pre-training dataset is naturally long-tailed, strongly encouraging a uniform prior leads to worse performance. However, we find a smaller penalty strength helps MSN to even outperform PMSN. This indicates that using a weak uniform prior can still be a reasonable choice when pre-training on long-tailed datasets.

Table 8: iNaturalist-2018 linear probing accuracy with full data

| Method | $K$ | $M$ | Overall | Head | Middle | Tail |
|---|---|---|---|---|---|---|
| *ViT-Small/16* | | | | | | |
| MSN ($\lambda = 1$) | 8142 | 3363 | **43.8** | **51.4** | **43.9** | **41.8** |
| MSN ($\lambda = 5$) | 8142 | 3123 | 42.3 | 49.6 | 42.5 | 40.4 |
| MSN ($\lambda = 15$) | 8142 | 1562 | 40.9 | 49.5 | 40.6 | 39.1 |
| PMSN ($\lambda = 1$) | 8142 | 2919 | 41.4 | 48.9 | 41.3 | **39.7** |
| PMSN ($\lambda = 5$) | 8142 | 3005 | **41.8** | **50.2** | **41.9** | **39.7** |
| PMSN ($\lambda = 15$) | 8142 | 2927 | 41.0 | 49.1 | 41.4 | 38.7 |

### A.4.3 TRANSFER LINEAR PROBING

We perform our transfer linear classification experiments on the standard suite of datasets used in self-supervised learning: Caltech101 (Li et al., 2022b), CIFAR10, CIFAR100 (Krizhevsky, 2009),

DTD (Cimpoi et al., 2014), Flowers (Nilsback & Zisserman, 2008), Food (Bossard et al., 2014), Pets (Parkhi et al., 2012) and SUN397 (Xiao et al., 2010). We follow the evaluation protocol from Ericsson et al. (2021); Chen et al. (2020) and train $L^2$-regularized linear classifiers. We select the regularization strength among a set of 45 values spaced linearly in the range $[-6, 5]$ in log-space and report the standard evaluation metric for each dataset.

### A.4.4 SINKHORN-KNOPP AND MEAN ENTROPY MAXIMIZATION HYPERPARAMETERS

For Sinkhorn-Knopp, we firstly use the vMF normalized version of iBOT and ablate over the number of iterations 1, 3, 5 and find that 3 iterations to work best. This choice for the number of iterations is in agreement with DINOv2 (Oquab et al., 2023). For both SK (iter=3) and mean entropy maximization and for each compute budget (2, 4 or 8 GPUs for 2 days) we ablate over the following hyperparameters:

- vMF normalization: True / False (Govindarajan et al., 2023)
- Teacher temperature:
  - $\tau = 0.04 \rightarrow 0.07$ (default in Zhou et al. (2022); Govindarajan et al. (2023))
  - $\tau = 0.05 \rightarrow 0.025$ (default in Ruan et al. (2023))

For SK(iter=3), we find smaller teacher temperatures to be beneficial as in Ruan et al. (2023) and using vMF normalization or not has marginal impact on the performance. For ME-MAX, we find that not using vMF normalization and a smaller teacher temperature leads to better performance.

### A.4.5 FINE-TUNING RECIPES

**ImageNet fine-tuning**: We fine-tune on the ImageNet dataset by following the fine-tuning recipe used in BeIT (Bao et al., 2022) and iBOT (Zhou et al., 2022), which is found to produce consistently good performance in reasonably fewer epochs compared to other fine-tuning recipes. We fine-tune ViT-Small and ViT-Base models for 200 and 100 epochs respectively and use a batch size of 1024. We use a layer-wise learning rate decay of 0.75 for ViT-Small and 0.65 for ViT-Base. We report the best performance achieved after considering 4 different learning rates: $\{8e-4, 9e-4, 1e-3, 2e-3\}$.

**iNaturalist-2018 fine-tuning**: We find the fine-tuning recipe of DeIT (Touvron et al., 2021) using a smaller learning rate and a larger number of epochs to work better for the iNaturalist-2018 dataset. This is similar to the transfer fine-tuning setup of iBOT (Zhou et al., 2022). We use a fine-tune both ViT-Small and ViT-Base models for 360 epochs using a batch size of 1024. We use learning rates of $5e-5$ and $7.5e-6$ for ViT-Small and ViT-Base respectively.

## A.5 ADDITIONAL RESULTS

### A.5.1 PARTIAL PROTOTYPE COLLAPSE IN MORE EXISTING MODELS

In addition to investigating partial prototype collapse in Table 1, we also investigate other self-supervised clustering methods that use a prototypical formulation such as EsViT (Li et al., 2022a) and SWaV (Caron et al., 2020). We demonstrate in Table 9 that partial prototype collapse also occurs in these methods. We observe that partial prototype collapse also occurs in methods using Resnet50 (He et al., 2016), ViL (Zhang et al., 2021) and CvT (Wu et al., 2021) backbones. Though we focus on ViT backbone models in this work, note that partial prototype collapse is not only limited to ViT backbones.

### A.5.2 ABLATION EXPERIMENT FOR KOLEO-PROTOTYPE REGULARIZATION STRENGTH

We conduct an ablation experiment to evaluate the impact of the regularization strength ($\lambda$) of the KoLeo-proto regularization term. We consider a 100 epoch iBOT-vMF pre-training using 8192 prototypes on the Imagenet dataset and evaluate $\lambda = \{0.02, 0.1, 0.5\}$. From Table 10, we find that too small $\lambda = 0.02$ is unable to fully utilize all the initialized prototypes. We observe improved performance and effective utilization of the prototypes using $\lambda = 0.1$ but do not observe further improvements from increasing $\lambda$ further. The main goal of this regularization is to effectively utilize the prototypes. We use the minimum regularization strength $\lambda = 0.1$ which is sufficient to achieve this in the experiments in this paper.

Table 9: Number of unique prototypes in existing models with $\epsilon = 0.025$ (pre-trained on ImageNet-1K)

| Backbone | Method | Initialized prototypes ($K$) | Unique prototypes ($M$) |
|---|---|---|---|
| Resnet50 | SWaV | 3000 | 1669 |
| Resnet50 | DINO | 60000 | 984 |
| Swin-Tiny/W=7 | EsViT | 65536 | 1157 |
| Swin-Base/W=14 | EsViT | 65536 | 4088 |
| ViL | EsViT | 65536 | 1741 |
| CvT | EsViT | 65536 | 1178 |

Table 10: Ablation experiment for KoLeo-proto regularization strength ($\lambda$)

| $\lambda$ | Initialized prototypes ($K$) | Unique prototypes ($M$) | kNN top-1 accuracy |
|---|---|---|---|
| 0.0 | 8192 | 1045 | 72.39 |
| 0.02 | 8192 | 4693 | 72.56 |
| 0.1 | 8192 | **8192** | **72.62** |
| 0.5 | 8192 | **8192** | **72.64** |

### A.5.3 IMAGENET PRE-TRAINING WITH A CNN BACKBONE

In order to explore an additional method and backbone combination, we consider the DINO method pre-training using a Resnet50 backbone. We base our pre-training settings on the hyperparameter configuration in the publicly available DINO codebase [2]. We use the vMF normalized version, use probability centering, 8192 prototypes and train for 100 epochs. In Table 11, we observe that the KoLeo-proto regularization mitigates the partial prototype collapse and achieves improved performance compared to the baseline and KoLeo-data regularization.

Table 11: ImageNet classification with full data (kNN, linear) using Resnet50 backbone model

| Method | Epochs | $M$ | kNN | Linear |
|---|---|---|---|---|
| DINO-vMF | 100 | 684 | 59.1 | 69.6 |
| DINO-vMF (kd) | 100 | 1373 | 59.8 | 70.4 |
| DINO-vMF (kp) | 100 | 8192 | **60.1** | **70.8** |

### A.5.4 COMPUTATIONAL ANALYSIS

The prototype layer in the self-supervised clustering methods that use a prototypical formulation noticeably contributes to the computational cost of training such methods. The weights associated with $K$ prototypes consists of a $K \times D$ matrix. Typically, the bottleneck dimension $D = 256$. The DINO models use a large $K = 65536$ and the prototype layer alone adds an additional 16M trainable parameters to the method. A batch of size $B$, results in the computation of probability distributions of size $B \times K$. For iBOT, which computes the probability distributions for all tokens resulting even larger set of probability distributions of size $B \times T \times K$. The number of prototypes in iBOT is set to $8192$ in the default configuration. Computing such large probability distributions involve heavy memory GPU usage and longer training times. For the default configurations of iBOT with ViT-S/16 backbone, we test the GPU memory use for different numbers of prototypes and batch sizes in the fp16 mode and report the results in Table 12. Consequently, effective utilization of prototypes can help in reducing the GPU memory required for training such models. For instance, effectively

---

[2]https://dl.fbaipublicfiles.com/dino/dino_resnet50_pretrain/args.txt

Table 12: Computational cost of training iBOT method with different number of prototypes

| Batch size ($B$) | Number of prototypes ($K$) | GPU memory (GB) |
|---|---|---|
| 64 | 1024 | 12.9 |
| 64 | 2048 | 13.7 |
| 64 | 4096 | 15.3 |
| 64 | 8192 | 18.6 |
| 64 | 10240 | 20.2 |
| 100 | 1024 | 19.7 |
| 100 | 2048 | 21.0 |
| 100 | 4096 | 23.5 |
| 100 | 8192 | 28.6 |
| 100 | 10240 | 31.1 |

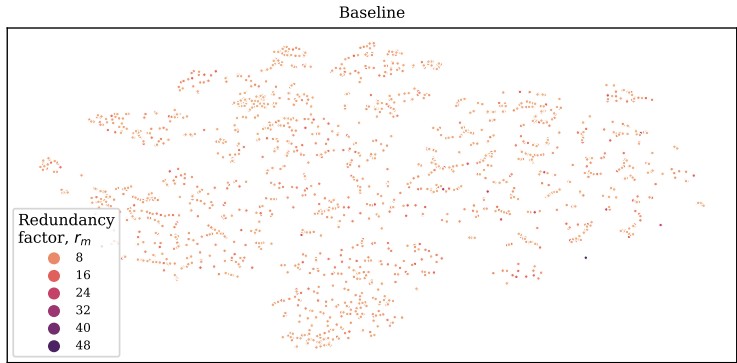

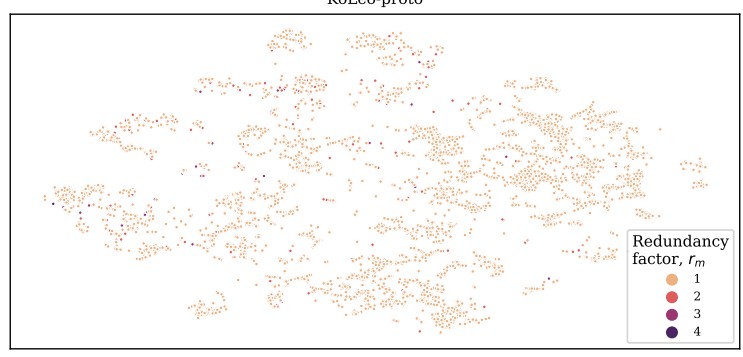

Figure 4: t-SNE plot of the $M$ unique prototypes learned by the baseline method and with KoLeo-proto regularization, colored by their redundancy factors $r_m$. There are fewer unique prototypes in the baseline ($M = 1806$), noticeable from their sparse spread in the plot. The baseline prototypes are impacted by partial prototype collapse, resulting in high redundancy factors. With KoLeo-proto regularization, the model learns more unique prototypes ($M = 7895$) with significantly smaller redundancy factors compared to the baseline. With KoLeo-proto regularization, the method learns diverse prototypes that are well spread over the latent space.

utilizing only 1024 prototypes is significantly cheaper than using only 1000 unique prototypes out of 8192 initialized prototypes.

## A.6 VISUAL EXPLANATIONS

In this section, we present a qualitative comparison of a model trained with and without KoLeo-proto regularization. We compare the iBOT-vMF baseline method based on the ViT-S/16 backbone

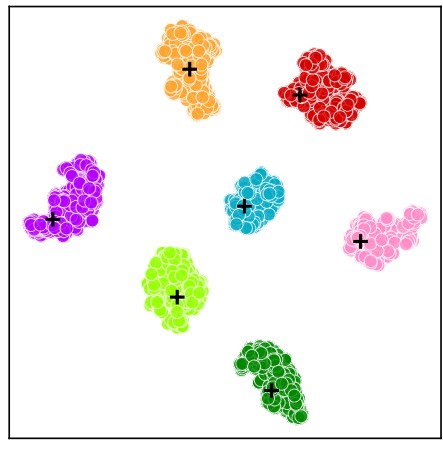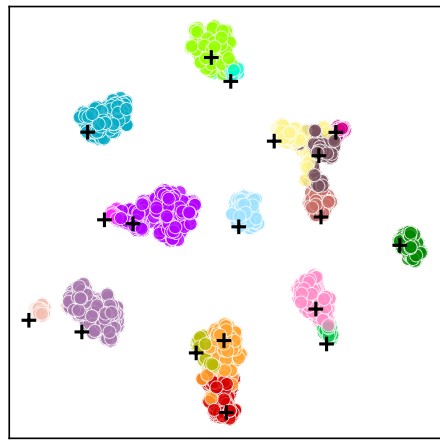

(a) iBOT-vMF baseline          (b) iBOT-vMF with KoLeo-proto

Figure 5: For the exact same set of images, the representations after the head (256 dimensional) are visualized using TSNE plots. The points are colored based on the latent class that they belong to and the corresponding prototypes are denoted using the + marker (the prototype markers are slightly shifted to prevent them from blocking some smaller clusters). The images belong to 7 latent classes in the iBOT-vMF baseline and the same images belong to 18 latent classes when the KoLeo-proto regularization is used. Partial prototype collapse in the baseline results in fewer unique prototypes and coarser clusters. KoLeo-proto regularization encourages diverse prototypes which leads to a more fine-grained clustering of the same data.

trained on the iNat18 dataset. We visualize the unique prototypes along with their redundancy factors in Figure 4 using t-SNE plots (Van der Maaten & Hinton, 2008). This illustrates the partial prototype collapse in the baseline and the impact of adding the KoLeo-proto regularization on the prototypes. KoLeo-proto regularization encourages diverse prototypes by spreading them out in the latent space, resulting in a higher number unique prototypes compared to the baseline. We visualize the representations corresponding to images that are assigned to a set of latent classes by the iBOT-vMF baseline in Figure 5a. In Figure 5b, we visualize the representations corresponding to the exact same images based on the iBOT-vMF model trained with KoLeo-proto regularization. We observe that the KoLeo-proto regularization encourages more fine-grained clusters compared to the baseline. In Figure 7 and Figure 6, we show a few example images belonging to the latent classes shown in Figure 5. Without KoLeo-proto regularization, only one coarse latent class is learned containing images of ducks. With KoLeo-proto regularization, this is further divided into three finer latent classes. This demonstrates that the model learns more informative representations which enable it to discriminate between these finer latent classes.

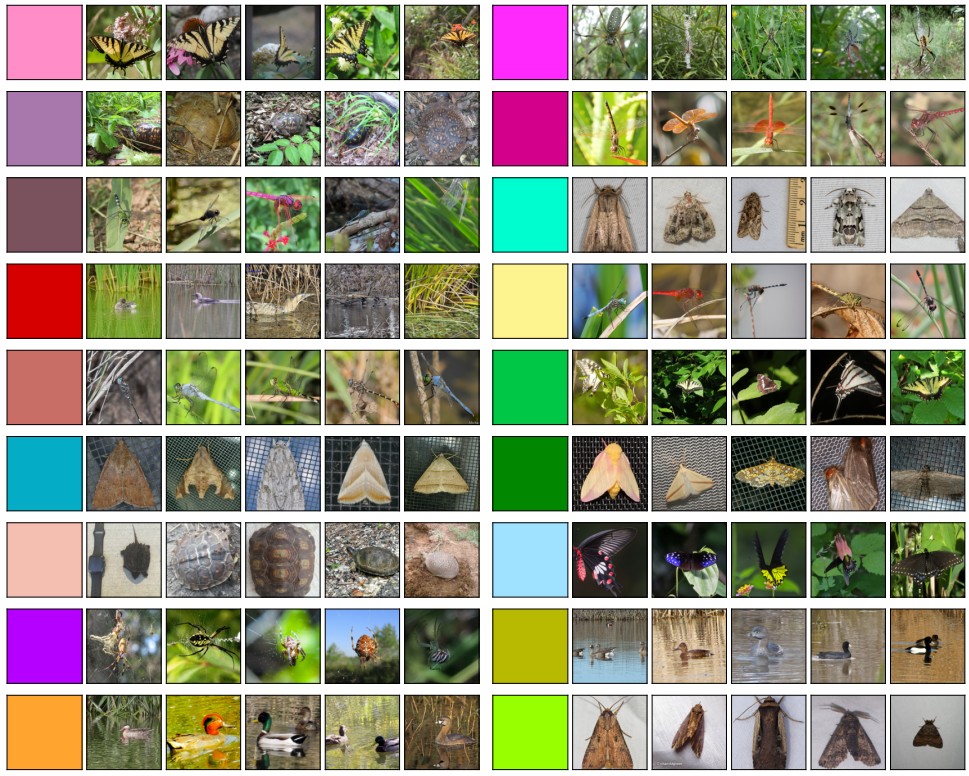

Figure 6: Sample images from the latent classes shown in 5b obtained from iBOT-vMF with KoLeo-proto regularization. Same colors are used to indicate the latent classes.

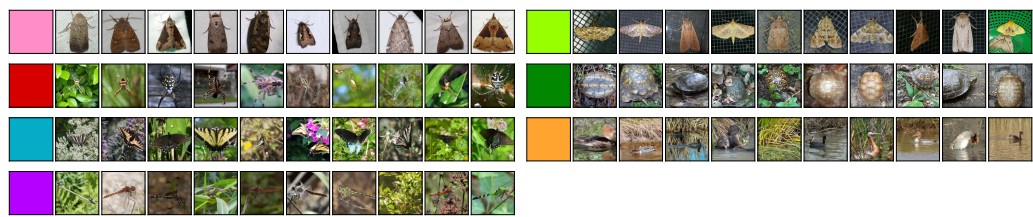

Figure 7: Sample images from the latent classes shown in 5a obtained from iBOT-vMF baseline method. Same colors are used to indicate the latent classes.

