# OpenReview forum: "On partial prototype collapse in clustering-based self-supervised learning"
_ICLR.cc/2024/Conference — Submitted to ICLR 2024_

### Official Review · Reviewer_ugdD · 2023-10-18

**Soundness:** 3 good
**Presentation:** 3 good
**Contribution:** 3 good
**Rating:** 6
**Confidence:** 4

**Summary:**

The authors found that a partial prototype collapse problem still exists in self-supervised learning methods.  So they proposed to regularize the diversity of prototypes, which can mitigate the partial prototype collapse problem. Experiments on several benchmarks demonstrate the benefit of this reguparization.

**Strengths:**

The authors defined a partial prototype collapse and demonstrate its occurrence in the DINO family of methods using different backbone models, and proposed KoLeo-proto regularization to prevent the partial prototype collapse. Overall, the paper is well organized and prepared.
1.  Self-supervised learning is very useful for representation learning, and the collapse problem seriously affecting the quality of features. The motivation of  solving a partial prototype collapse problem is soundness.
2. The experiments are comprehensive, including imagenet classification, transfer learning, long-tail dataset, and sensitivity analysis.

**Weaknesses:**

1. Is there any theoretical results to guarantee that minimizing $L_{KP}$ will make the prototypes to  be uniformly distributed in the latent space?  Why not just minimizing $-\sum_{i}\sum_{j}||\mu_{i}-\mu_{j}||^{2}$?
2. Overall, the improvement is small when compared with the baseline iBOT-vMF. For example, the classification accuracy of imagenet in Table 2 are very close for iBOT-vMF and  iBOT-vMF (kp).
3. The authors should give more quantitative or qualitative results to demonstrate how the proposed KP regularization can solve the partial prototype collapse problem, e.g. visualization.

**Questions:**

1. Is there any theoretical results to guarantee that minimizing $L_{KP}$ will make the prototypes to  be uniformly distributed in the latent space?  Why not just minimizing $-\sum_{i}\sum_{j}||\mu_{i}-\mu_{j}||^{2}$?
2. Overall, the improvement is small when compared with the baseline iBOT-vMF. For example, the classification accuracy of imagenet in Table 2 are very close for iBOT-vMF and  iBOT-vMF (kp).
3. The authors should give more quantitative or qualitative results to demonstrate how the proposed KP regularization can solve the partial prototype collapse problem, e.g. visualization.

---

> ### Author Response · Authors · 2023-11-19
> **Response to reviewer ugdD**
>
> We thank the reviewer for the insightful feedback and valuable comments. We appreciate that the reviewer recognizes the importance of the partial prototype collapse problem and for commenting on our overall presentation and experiments. We address the weaknesses and questions raised by the reviewer below.
>
> > W1. Is there any theoretical results to guarantee that minimizing L_KP will make the prototypes to be uniformly distributed in the latent space? Why not just minimizing $- \sum_i \sum_j || \mu_i - \mu_j ||^2$ ?
>
> The objective that you suggested was one of our first thoughts as well but we realized that this can be reduced to something trivial. Since $\mu_i$ and $\mu_j$ are unit vectors, $\sum_i \sum_j || \mu_i - \mu_j ||^2$ = $\sum_i \sum_j 2 - 2 \langle \mu_i, \mu_j \rangle$ = $2 - 2 \langle \sum_i \mu_i, \sum_j \mu_j \rangle$ = $2 - 2 || \sum_i \mu_i ||^2$.
>
> The objective reduces to minimizing the $L^2$-norm of the sum of the prototypes. This cannot solve the problem of partial prototype collapse. Consider a simple example where K/2 prototypes are equal to $\mu$ and the other K/2 prototypes are equal to $- \mu$. This will minimize the above objective perfectly but learns only 2 unique prototype vectors.
>
> For detailed theoretical results related to optimizing the KoLeo objective, kindly refer to [1]. The KoLeo objective is an estimate of the differential entropy. A uniform distribution of prototypes in the latent space results in maximum differential entropy.
>
> [1] Sablayrolles, Alexandre, et al. "Spreading vectors for similarity search." ICLR 2019.
>
> > W2. Overall, the improvement is small when compared with the baseline iBOT-vMF. For example, the classification accuracy of imagenet in Table 2 are very close for iBOT-vMF and iBOT-vMF (kp).
>
> Our secondary contribution of mitigating partial prototype collapse is aimed at effectively utilizing the prototypes which we manage to achieve. We study the downstream impact of effectively utilizing all the prototypes - we study uniform and long-tailed datasets under different tasks to understand both the advantages and limitations that arise from this. In the experiments reported in Table 2, we use the same number of initialized prototypes (K=8192) as the baseline. We do this to measure the performance improvement that can be obtained only from the effective utilization of the prototypes and not from making use of additional prototypes (which also changes the memory footprint of the model compared to the baseline; see Table 12 in revised paper for details on memory costs). As we note in section 6.3, we observe that the Imagenet kNN top-1 accuracy continues to improve as the number of prototypes are increased for iBOT-vMF with KoLeo-proto regularization, which is not the case with the baseline. The Imagenet classification performance with our proposed regularization can be improved further by scaling the number of prototypes. We do not investigate this for the longer experiments due to the significantly higher computational cost required for such experiments.
>
> > W3. The authors should give more quantitative or qualitative results to demonstrate how the proposed KP regularization can solve the partial prototype collapse problem, e.g. visualization.
>
> The goal of adding the KoLeo-prototype regularization is to mitigate the partial prototype collapse issue. We show in Figure 3a that our proposed regularization is able to achieve this goal, by comparing the number of unique prototypes learned. Through several evaluation experiments we further quantify the downstream impact of this regularization and how it varies for uniform and long-tailed pre-training datasets. We have now added section A.6 in the revised paper to provide an intuitive visual explanation of the partial prototype collapse problem and how the KoLeo-proto regularization impacts the prototypes and the clusters learned by the method.

---

> > ### Comment · Reviewer_ugdD · 2023-11-22
> > **Response**
> >
> > Thanks for your responses. In my opinion, this article is good, but not considered excellent. So I will keep my rating.

---

### Official Review · Reviewer_fvFf · 2023-10-30

**Soundness:** 2 fair
**Presentation:** 3 good
**Contribution:** 2 fair
**Rating:** 3
**Confidence:** 5

**Summary:**

This work reveals a partial prototype collapse problem which leads to significant redundancies in the prototypes. Such prototype redundancies serve as shortcuts for the method to achieve a marginal latent class distribution that matches the prescribed prior distribution. By encouraging the model to use diverse prototypes, the partial prototype collapse can be mitigated, which is especially beneficial when pre-training on a long-tailed fine-grained dataset. Effective utilization of the prototypes produces fine-grained clusters, encourages more informative representations, and may reduce computational expenses.

**Strengths:**

1. The paper is easy to follow with good writing, well organization, and comprehensive literature review.
2. The analysis of regularizations in the DINO family of SSL methods is helpful to grasp the principle of the proposed method.
3. The definition of partial prototype collapse is clear and reasonable.

**Weaknesses:**

1. The title of this work seems to explore the partial prototype collapse for all clustering-based SSL methods. However, without any clarification, all explorations and discussions are focused on the DINO family of SSL methods by default.
2. The last sentence of the Sec. Conlusion claims the superior learning efficiency of the proposed method. However, there is no experiment to prove that.
3. Although several experiments justify the efficacy of the proposed method on long-tailed fine-grained datasets and few-shot settings, there is no convincing explanations.
4. The proposed method is implicitly built on an assumption that improving the uniformity of the representation space while keep its alignment will produce a better sample distribution. However, without superior distribution priors, the trials of the proposed methods report limited performance improvements on standard SSL evaluations.

**Questions:**

See the weaknesses.

---

> ### Author Response · Authors · 2023-11-19
> **Response to reviewer fvFf (part 1/2)**
>
> We thank the reviewer for the useful feedback. We appreciate that the reviewer found our paper to be well presented in terms of writing and organization. We consider our identification and definition of the partial prototype collapse to be one of the key contributions of our work and we appreciate that the reviewer considers it as one of the strengths. Our secondary contribution is to propose KoLeo-proto regularization as a way to mitigate partial prototype collapse. We study the downstream impact of effectively utilizing the prototypes in different scenarios and report both the advantages and the limitations.
>
> > W1. The title of this work seems to explore the partial prototype collapse for all clustering-based SSL methods. However, without any clarification, all explorations and discussions are focused on the DINO family of SSL methods by default.
>
> We agree that there is a slight disconnect between the paper title and the experiments in the paper. Note, however, that we consider the main contribution of the paper to be the definition and identification of the partial prototype collapse problem, and this is what the title refers to. This contribution applies to self-supervised clustering methods that use a prototypical formulation which includes SwAV, DINO, EsViT, MSN and iBOT. The DINO family of methods using ViT backbones is the predominant recipe among self-supervised clustering methods. This is our motivation for focusing on the DINO family in the paper. We have now furthermore investigated the SwAV and EsViT pre-trained models using various backbones and find that they also suffer from a partial prototype collapse (added to Table 9 in revised paper). For the SwAV Resnet50 backbone model, only 1669 prototypes are unique out of the 3000 initialized prototypes.
>
> For the investigation of the effect of KoLeo prototype regularization (a secondary contribution) it is true that we focus on DINO and iBOT. The DINO family of methods is the prominent approach to self-supervised clustering in recent times. Several methods such as MSN, EsViT, iBOT are based on the DINO objective. The iBOT formulation is a strong recent baseline among the DINO family of methods and also used as the foundation for DINOv2. The vMF version is chosen because it  stabilizes the iBOT training [1] (and provides improved performance). This is our motivation for choosing iBOT-vMF as the baseline for evaluating our second contribution of mitigating partial prototype collapse through KoLeo-prototype regularization.
>
> [1] Govindarajan, Hariprasath, et al. "DINO as a von Mises-Fisher mixture model." ICLR 2023.
>
> > W2. The last sentence of the Sec. Conlusion claims the superior learning efficiency of the proposed method. However, there is no experiment to prove that.
>
> There appears to be a misunderstanding regarding what we mean in the last two sentences of the conclusion section. We merely point out that the computational cost of training the DINO family of methods depends (in a non-negligible way) on the number of latent classes, K. Computing probability distributions over K latent classes involves a prototype layer of size $K \times 256$. The probability distributions in DINO are of the size $B \times K$ and in iBOT they are of the size $B \times T \times K$, where T is the number of tokens (T = 197 in the default configuration). The choice of K has a notable impact on the training times and memory use.
>
> We find that around 1000 unique prototypes are used when the number of initialized prototypes are 2048, 4096, 8192 or 10240. For a batch size of 100, using 2048 vs 10240 prototypes consumes GPU memory of 21 GB vs 31.1 Gb respectively. We quantify this for different batch sizes and number of prototypes in section A.5.4 of revised paper. Considering the computational cost associated with using more prototypes, it is important to effectively utilize the initialized prototypes. This is made possible by our proposed regularization, which itself adds negligible computational cost in terms of memory and time.
>
> When pre-training on another dataset, if a smaller number of prototypes are sufficient, then initializing the model with fewer prototypes and effectively utilizing them will bring computational gains compared to setting the number of prototypes to a larger number along with the occurrence of partial prototype collapse.
>
> > W3. Although several experiments justify the efficacy of the proposed method on long-tailed fine-grained datasets and few-shot settings, there is no convincing explanations.
>
> Our intuitive explanation is that learning fine-grained clusters encourages the model to learn more informative–but also more low-level and dataset specific–features that are able to discriminate between the fine-grained clusters. This is intuitively useful for a fine-grained classification task, as well as for few-shot learning. However, some of these additional features may not generalize well to other datasets, thus hurting transfer performance.

---

> ### Author Response · Authors · 2023-11-19
> **Response to reviewer fvFf (part 2/2)**
>
> > W4. The proposed method is implicitly built on an assumption that improving the uniformity of the representation space while keep its alignment will produce a better sample distribution. However, without superior distribution priors, the trials of the proposed methods report limited performance improvements on standard SSL evaluations.
>
> Sorry, we do not fully understand what the reviewer meant by “The proposed method is implicitly built on an assumption that improving the uniformity of the representation space while keep its alignment will produce a better sample distribution”. We can provide a better explanation if the reviewer can elaborate this.
>
> We discuss two forms of regularization with different purposes - MLCD regularization and prototype regularization. MLCD regularization is used to prevent complete representation collapse. As a consequence of partial prototype collapse (PPC), the method can cheat and satisfy the MLCD constraint *regardless of the choice of prior and MLCD regularization technique* by allowing the prototypes to partially collapse. This results in fewer distinct latent classes for the self-supervision task. The prototype regularization is aimed at mitigating partial prototype collapse and learning diverse prototypes. The distribution of representations in the DINO family of methods is shown to correspond to that of a mixture of von Mises-Fisher components [1].
>
> We study the downstream impact of effectively utilizing all the prototypes - we study uniform and long-tailed datasets under different tasks to understand both the advantages and limitations that arise from this. In the experiments reported in Table 2, we use the same number of initialized prototypes (K=8192) as the baseline. We do this to measure the performance improvement that can be obtained only from the effective utilization of the prototypes and not from making use of additional prototypes (which also changes the memory footprint of the model compared to the baseline; see Table 12 in revised paper for details on memory costs). As we note in section 6.3, we observe that the Imagenet kNN top-1 accuracy continues to improve as the number of prototypes are increased for iBOT-vMF with KoLeo-proto regularization, which is not the case for the baseline. The Imagenet classification performance with our proposed regularization can be improved further by scaling the number of prototypes. We do not investigate this for the longer experiments due to the significantly higher computational cost required for such experiments.
>
> [1] Govindarajan, Hariprasath, et al. "DINO as a von Mises-Fisher mixture model." ICLR 2023.

---

> ### Author Response · Authors · 2023-11-23
> **Discussion period ending soon**
>
> We understand that it has only been a few days since our response. We would like to kindly remind the reviewer that the discussion period is going to end soon. We sincerely appreciate if the reviewer takes our responses into consideration.

---

### Official Review · Reviewer_4tkT · 2023-10-31

**Soundness:** 3 good
**Presentation:** 2 fair
**Contribution:** 2 fair
**Rating:** 6
**Confidence:** 4

**Summary:**

This paper investigates the issue of Partial Prototype Collapse (PPC) in self-supervised learning (SSL), which can lead to a significant decrease in classification performance. Additionally, the study introduces a novel prototype-based regularization strategy and experimental results confirm the effectiveness of this strategy in mitigating the PPC phenomenon.

**Strengths:**

1. This study marks the first to propose the issue of Partial Prototype Collapse as an inherent challenge in the field of self-supervised learning (SSL). Furthermore, it extensively delves into the existing SSL methods in the Introduction section.

2. The proposed method is simple yet effective. The overall solution is technique sounds and might become a new baseline of SSL.

3. The paper's structure is well-organized, and it provides a relatively clear description of the method.

4. The claims made in the paper are substantiated by the results and findings obtained through the conducted experiments.

**Weaknesses:**

1.  The employed objective function is rather conventional, which may limit the overall innovation of the study.

2. The experiments in the paper mainly compare the proposed strategy with the baselines, without a direct comparison with the existing regularization techniques.

3. The impact of regularization paramete $\lambda$ on the effectiveness of the proposed regularization method may not have been thoroughly explored in the paper.

4. The paper heavily relies on textual descriptions, with limited use of graphics to explain the issues or algorithms, which might pose difficulties for readers.

**Questions:**

1. How is the number of prototypes determined for different datasets?

2. How does the proposed regularization strategy perform in terms of robustness under different hyperparameters?

3. The paper claims that the proposed method is more effective in long-tail classification tasks. Does this suggest that long-tail distributions are more likely to lead to Partial Prototype Collapse?

4. Employing visuals to illustrate Partial Prototype Collapse could enhance readers' comprehension.

5. Is it necessary to determine the initial number of prototypes before the algorithm is executed?

6. Some details of the experimental setup were not clarified. For instance, what optimizer was used?

7. This paper only utilizes various ViT-based architectures, does the Partial Prototype Collapse also observed in CNN-based architectures?

---

> ### Author Response · Authors · 2023-11-19
> **Response to reviewer 4tkT (part 1/2)**
>
> We thank the reviewer for the insightful feedback and valuable comments. We appreciate that the reviewer recognizes our identification of the Partial Prototype Collapse in self-supervised learning methods based on the clustering task. We address the weaknesses and questions raised by the reviewer below.
>
> > W1. The employed objective function is rather conventional, which may limit the overall innovation of the study.
>
> We agree that the KoLeo objective for spreading vectors in a latent space [1] is not novel. However, we would like to point out that its usage in the context of learning diverse prototypes is new. Though this solution might be derivative, we believe that identifying and defining the partial prototype collapse issue is a key contribution that is of importance to this family of methods.
>
> [1] Sablayrolles, Alexandre, et al. "Spreading vectors for similarity search." ICLR 2019.
>
> > W2. The experiments in the paper mainly compare the proposed strategy with the baselines, without a direct comparison with the existing regularization techniques.
>
> The complete representation collapse is a known problem in self-supervised learning where the representations of all data samples converge to the same vector. This is addressed using various architectural tricks and regularization methods. In clustering-based self-supervised methods, the common way to prevent complete collapse is through MLCD regularization. We unify several existing MLCD regularization techniques in section 3 of the paper and follow-up with comparative experiments in section 6.1. However, the partial prototype collapse is a new problem that we have found in this family of self-supervised learning methods. Since this problem was not known previously, there are no existing regularization techniques to address this specific issue. Our proposed KoLeo-proto regularization can be used in conjunction with different MLCD regularization techniques.
>
> > W3. The impact of regularization parameter \lambda on the effectiveness of the proposed regularization method may not have been thoroughly explored in the paper.
>
> We apologize for not including this ablation experiment in the earlier version of the paper. We conducted an ablation experiment on the regularization parameter and found that $\lambda=0.1$ is sufficient to utilize all the prototypes and increasing the strength further does not have any qualitative or quantitative improvements. The ablation experiment is added to section A.5.2 in the revised paper.
>
> > W4. The paper heavily relies on textual descriptions, with limited use of graphics to explain the issues or algorithms, which might pose difficulties for readers. AND
>
> > Q4. Employing visuals to illustrate Partial Prototype Collapse could enhance readers' comprehension.
>
> We added section A.6 in the appendix to provide an intuitive explanations of our results:
> - We provide t-SNE plots of all the learned prototypes to illustrate how the prototypes are impacted by partial prototype collapse and how KoLeo-proto regularization is able to spread the prototypes and hence, learn more diverse prototypes.
> - We show t-SNE plots of the same subset of data points based on representations from the baseline iBOT-vMF method and after adding KoLeo-proto regularization (from the iNat18 experiment with ViT-S/16 backbone). We demonstrate that the KoLeo-proto regularization leads to more fine-grained cluster structure for the same data when compared to the baseline. We complement this by presenting some example images belonging to these clusters.
>
> We hope that these newly added illustrations complement the textual explanations in the main paper and improve the readability of the paper.
>
> > Q1. How is the number of prototypes determined for different datasets?
>
> Typically, the number of prototypes are determined through ablation experiments in prior literature. However, because of the partial prototype collapse, this hyperparameter previously did not perform its intended purpose of controlling the number of clusters learned by the method. Through KoLeo-proto regularization, we enable effective utilization of the initialized prototypes. Our work makes this hyperparameter relevant and sheds light on the impact of this choice. In our experiments, we use the same number of initialized prototypes as the baselines in order to only demonstrate the impact of their effective utilization (exact numbers used in our experiments are listed in the appendix section A.4.1).
>
> Selecting the number of prototypes for a new unfamiliar dataset is a common challenge for this family of methods. One could design methods that can also learn the optimal number of clusters for a dataset. This is a difficult problem since we are learning representations and the clustering model of the representations simultaneously. We consider this as an interesting direction for future research.

---

> ### Author Response · Authors · 2023-11-19
> **Response to reviewer 4tkT (part 2/2)**
>
> > Q2. How does the proposed regularization strategy perform in terms of robustness under different hyperparameters?
>
> We mainly experiment with adding the KoLeo-proto regularization to the iBOT-vMF baseline method (with different ViT backbones). Additionally, we have now investigated DINO using a Resnet50 backbone. We also explore using different types of pre-training datasets - uniform and long-tailed. We use the default hyperparameter configurations for all the models, which have been identified by the original works after extensive experimentation. However, note that these hyperparameter configurations vary between ViT-S/16, ViT-B/16 and Resnet50 backbones. This includes differences in multicrop settings, epochs, learning rate, temperature schedule and weight decay. We find the proposed regularization to be robust over these hyperparameter choices.
>
> > Q3. The paper claims that the proposed method is more effective in long-tail classification tasks. Does this suggest that long-tail distributions are more likely to lead to Partial Prototype Collapse?
>
> In our experiments, we observe partial prototype collapse in both uniform and long-tailed datasets, not suggesting that it is more likely in the latter. As brought up by reviewer hqpg, a minority collapse problem is identified in supervised long-tail classification [2]. However, the partial prototype collapse occurs in the self-supervised context among latent classes that are learned by the model. For a long-tailed dataset, the partial prototype collapse results in learning more coarse-grained latent classes where the number of unique prototypes are less than the number of classes. Note that this can also occur in fine-grained datasets with many classes. When models use fewer unique prototypes than the number of classes, the learned clusters are likely to have merged several of the fine-grained classes (or classes from the long tail). This can lead to less informative representations which cannot discriminate between fine-grained classes. This can disproportionately affect long-tailed or fine-grained classification datasets.
>
> [2] Fang, Cong, et al. "Exploring deep neural networks via layer-peeled model: Minority collapse in imbalanced training." Proceedings of the National Academy of Sciences, 2021
>
> > Q5. Is it necessary to determine the initial number of prototypes before the algorithm is executed?
>
> For existing methods in the DINO family, it is necessary to make this hyperparameter choice before training the method. It can be of interest as stated above to explore ways to avoid this. Previous works have relied on ablation experiments to select the number of prototypes but they were affected by partial prototype collapse. Our work sheds light on the importance and impact of this choice.
>
> > Q6. Some details of the experimental setup were not clarified. For instance, what optimizer was used?
>
> We use the adamw optimizer for all the model trainings. The complete experimental setup including the optimizer settings are provided in section A.4 of the appendix.
>
> > Q7. This paper only utilizes various ViT-based architectures, does the Partial Prototype Collapse also observed in CNN-based architectures?
>
> Yes, this is also observed in the CNN-based architectures, for instance, we observe partial prototype collapse in the DINO and SWaV models which use the Resnet50 backbone. We also investigate the EsViT models pre-trained using other Transformer variants such as ViL and CvT. This additional investigation is presented in section A.5.1 of the revised paper. We also explore the proposed KoLeo-proto regularization with the DINO method and a Resnet50 backbone and observe improved prototype utilization and performance (see section A.5.3 in the revised paper).

---

> > ### Comment · Reviewer_4tkT · 2023-11-22
> >
> > The responses answer all of my questions. The proposed algorithm has its own merits. Some of my concerns are unaddressed regarding the verification on broader techniques of clustering for prototype generation and in-depth analysis. I think it's a boarderline paper.

---

### Official Review · Reviewer_hqpg · 2023-11-01

**Soundness:** 3 good
**Presentation:** 3 good
**Contribution:** 3 good
**Rating:** 6
**Confidence:** 4

**Summary:**

This paper underlines a prevailing phenomenon of previous self-supervised learning works:the regularization of marginal latent class distribution to a specific prior distribution, often leading to partial prototype collapse. Based on this observation, this paper proposes a regularization loss that maximizes the differential entropy of the prototype vectors, aiming to enhance cluster diversity. The experiments are focused on improving the performance on long-tailed fine-grained datasets, specifically demonstrated on the linear probing accuracy on iNaturalist-2018.

**Strengths:**

1.	The recognition of partial prototype collapse illuminates a previously overlooked phenomenon for SSL.
2.	This paper proposes an interesting term to quantitatively assess the uniqueness of prototypes and pinpoints the shared characteristics of previous SSL works, namely the mismatch between initialized prototypes and unique prototypes.
3.	The writing is clear, offering both numerical experiments for elucidation and theoretical deductions for integrating previous works cohesively.
4.	The proposed method brings some noticeable improvement on its baseline, tackling the long-tailed scenario with minor adjustments to the method design.

**Weaknesses:**

1.	The proposed method mainly works on long-tailed datasets, however, its motivation does not rigorously align with its goal. Although the authors mention that previous works encourage MLCD to match a prescribed uniformed prior distribution, why could KoLeo-proto regularization move away from such a conventional prior? More elaborations are needed to shed light on how the KoLeo-proto regularization could help with the long-tailed problem.

2.	The KoLeo-proto regularization mainly aims to improve the diversity of the learned clusters while reducing partial prototype collapse. Another stream of study [R1], which explores the neural collapse phenomenon that also scatters the prototypes into an ETF geometric structure, also proves to be effective for imbalanced learning. Are there any similar spirits between this KoLeo-proto regularization and the neural-collapse-inspired design?
[R1] Inducing Neural Collapse in Imbalanced Learning: Do We Really Need a Learnable Classifier at the End of Deep Neural Network, NeurIPS, 2022.

3.	The KoLeo-proto regularization does not seem to improve the transfer performance (-0.7% on ViT-base and +0.1% on ViT-small ), as pointed out by the authors in Table 3. The authors explain that it is a common trade-off also existing in other works whose performance thrives in few-shot scenarios but falls short in transfer learning. Unfortunately, there are no further explanations for why this trade-off exists. Also, it seems that KoLeo-proto generally gains negligible improvement on finetuning tasks, as also indicated in Table2, 4. It would be better to elaborate on the results.

**Questions:**

This paper mainly takes iBoT-vMF as its baseline. Considering the simple and elegant form of KoLeo-proto regularization, would it also benefit other baselines either in qualitative (i.e. the uniqueness of prototypes) or quantitative (i.e. the performance) ways?

---

> ### Author Response · Authors · 2023-11-19
> **Response to reviewer hqpg (part 1/3)**
>
> We thank the reviewer for the constructive feedback and valuable comments. We address the weaknesses and questions raised by the reviewer below.
>
> > W1. The proposed method mainly works on long-tailed datasets, however, its motivation does not rigorously align with its goal. Although the authors mention that previous works encourage MLCD to match a prescribed uniformed prior distribution, why could KoLeo-proto regularization move away from such a conventional prior? More elaborations are needed to shed light on how the KoLeo-proto regularization could help with the long-tailed problem.
>
> We do not in fact move away from encouraging the MLCD to match a prescribed prior distribution. This could be a uniform (considered by most previous work) or non-uniform (considered by, e.g., PMSN). As a consequence of partial prototype collapse (PPC), the method can cheat and satisfy the MLCD constraint *regardless of the choice of prior and MLCD regularization technique* by allowing the prototypes to partially collapse. KoLeo-proto regularization is a way to prevent this type of collapse, removing the possibility of “cheating” in this way, which makes the MLCD regularization more effective. This is true for any choice of prior and MLCD regularization technique. It is true that we focus on the common setting with a uniform prior, together with probability centering as MLCD regularization, but the PPC issue is not limited to this case. We find in Table 8 that by choosing a better KL penalty strength, we can also achieve better performance on long-tailed data by encouraging MLCD to match a uniform prior rather than a long-tailed prior (comparing MSN and PMSN). This is our motivation for focusing on the uniform prior in our experiments. Also, note in Table 4 that both MSN and PMSN learn ~3000 unique prototypes out of the 8142 initialized prototypes, despite using different priors (uniform in MSN and long-tailed in PMSN).
>
> We provide a simple example to explain how a model can “cheat” through partial prototype collapse to match the MLCD to a prescribed uniform prior. Consider a dataset of size D and 4 initialized prototypes. Then, consider the following sets of probability distributions corresponding to the D samples when the number of unique prototypes learned by the model are 4 and 2:
>
> 1. 4 unique prototypes: $[v_1, v_2, v_3, v_4]$
>     - D/4 samples: $[1.0, 0.0, 0.0, 0.0]$
>     - D/4 samples: $[0.0, 1.0, 0.0, 0.0]$
>     - D/4 samples: $[0.0, 0.0, 1.0, 0.0]$
>     - D/4 samples: $[0.0, 0.0, 0.0, 1.0]$
> 2. 2 unique prototypes: $[v_1, v_1, v_2, v_2]$
>     - D/2 samples: $[0.5, 0.5, 0.0, 0.0]$
>     - D/2 samples: $[0.0, 0.0, 0.5, 0.5]$
>
> Both configurations of probability distributions achieve a uniform MLCD. However, they use different numbers of unique prototypes. Classifying the data samples into more latent classes is a harder task, as is the case in the first configuration. This encourages the model to learn more informative representations to discriminate between the fine-grained latent classes. We achieve this using the KoLeo-proto regularization that ensures effective utilization of the prototypes.
>
> For a long-tailed dataset, the partial prototype collapse results in learning more coarse-grained latent classes where the number of unique prototypes are less than the number of classes. We confirm this to be the case using the iNaturalist-2018 dataset (see Table 4; column M in the paper). When models use fewer unique prototypes than the number of classes, the learned clusters are likely to have merged several of the fine-grained classes (typically, classes from the long tail). This is mitigated when the prototypes are effectively utilized, leading to more diverse clusters and hence, more informative representations.

---

> ### Author Response · Authors · 2023-11-19
> **Response to reviewer hqpg (part 2/3)**
>
> > W2. The KoLeo-proto regularization mainly aims to improve the diversity of the learned clusters while reducing partial prototype collapse. Another stream of study [R1], which explores the neural collapse phenomenon that also scatters the prototypes into an ETF geometric structure, also proves to be effective for imbalanced learning. Are there any similar spirits between this KoLeo-proto regularization and the neural-collapse-inspired design? [R1] Inducing Neural Collapse in Imbalanced Learning: Do We Really Need a Learnable Classifier at the End of Deep Neural Network, NeurIPS, 2022.
>
> We thank the reviewer for bringing this work [1] to our notice, where the last linear layer of a supervised classifier is replaced with prototypes initialized using an ETF geometric structure which are not trained. If we apply the KoLeo-proto regularization with a large strength, we would learn a similar ETF structure. However, this makes the implicit assumption that all classes are equally similar/dissimilar to one another, which is a strong assumption. Such a hard constraint may not be a suitable choice in the self-supervised setting, where the objective is to learn good representations. It should be possible for semantically similar clusters to be located nearer in the latent space instead of being separated by a fixed distance. Hence, we argue that a softer penalty through the KoLeo-proto regularization can suit the self-supervised learning scenario better than using a fixed ETF.
>
> Our explanation above (in response to the previous question) for why effective utilization of the prototypes is beneficial for long-tailed datasets is partly related to the minority collapse issue explored in [2]. In the supervised case, [2] finds that the minority classes were merged with one another, thus increasing the similarity between representations of different minority classes. In the self-supervised learning case, since we do not use class labels during pre-training, this happens at a latent class level and the minority class samples can be merged with samples from any other class (head/mid/tail). Also note that PPC is not just limited to long-tailed data (also shown to occur in ImageNet).
>
> [1] Yang, Yibo, et al. "Inducing Neural Collapse in Imbalanced Learning: Do We Really Need a Learnable Classifier at the End of Deep Neural Network?." NeurIPS, 2022.
>
> [2] Fang, Cong, et al. "Exploring deep neural networks via layer-peeled model: Minority collapse in imbalanced training." Proceedings of the National Academy of Sciences, 2021.
>
> > W3. The KoLeo-proto regularization does not seem to improve the transfer performance (-0.7% on ViT-base and +0.1% on ViT-small ), as pointed out by the authors in Table 3. The authors explain that it is a common trade-off also existing in other works whose performance thrives in few-shot scenarios but falls short in transfer learning. Unfortunately, there are no further explanations for why this trade-off exists. Also, it seems that KoLeo-proto generally gains negligible improvement on finetuning tasks, as also indicated in Table2, 4. It would be better to elaborate on the results.
>
> This trade-off is a more generic issue not specific to our work, and although we found it interesting to observe that the tradeoff is present also in the current work, we consider a more in-depth investigation  to be out-of-scope for this paper. We have not investigated this in sufficient depth to provide a clear explanation. A possible reasoning for reduced transfer performance is that the features learned to discriminate between the fine-grained clusters (which is encouraged by KoLeo-proto regularization) become more  specific to the pre-training dataset. These features may not generalize well to other datasets, resulting in reduced transfer performance.

---

> ### Author Response · Authors · 2023-11-19
> **Response to reviewer hqpg (part 3/3)**
>
> > Q1. This paper mainly takes iBoT-vMF as its baseline. Considering the simple and elegant form of KoLeo-proto regularization, would it also benefit other baselines either in qualitative (i.e. the uniqueness of prototypes) or quantitative (i.e. the performance) ways?
>
> Yes, the KoLeo-proto regularization can be easily added to several prototypical formulations - which is the case for SWaV and the DINO family of methods including EsViT, iBOT and MSN. We focus our experiments on iBOT which is a recent state-of-the-art method in the DINO family. The vMF extension stabilizes the iBOT training and provides a further performance boost [3]. Even the most recent DINOv2 work builds on top of iBOT. This is our motivation for our choice of iBOT-vMF as the baseline. SWaV, EsViT, iBOT and MSN also suffer from a partial prototype collapse (see Table 1 for iBOT, MSN. See below for EsViT and SWaV; added to Table 9 in the revised paper) and this can be mitigated using the KoLeo-proto regularization. We conducted an additional experiment exploring KoLeo-proto regularization on DINO and used the Resnet50 backbone (to explore a CNN backbone, complementing our previous ViT experiments). The results are in agreement with the results observed for ViT backbones (see below; added to section A.5.3 in the appendix).
>
> [3] Govindarajan, Hariprasath, et al. "DINO as a von Mises-Fisher mixture model." ICLR 2023.
>
> **Number of unique prototypes in existing models**
>
> | Backbone       | Method | Initialized prototypes (K) | Unique prototypes (M) |
> |----------------|--------|----------------------------|-----------------------|
> | Resnet50       | SWaV   | 3000                       | 1669                  |
> | Resnet50       | DINO   | 60000                      | 984                   |
> | Swin-Tiny/W=7  | EsViT  | 65536                      | 1157                  |
> | Swin-Base/W=14 | EsViT  | 65536                      | 4088                  |
> | ViL            | EsViT  | 65536                      | 1741                  |
> | CvT            | EsViT  | 65536                      | 1178                  |
>
> **ImageNet classification with full data (kNN, linear) using DINO - Resnet50 backbone model**
>
> | Method        | Epochs | Initialized prototypes (K) | Unique prototypes (M) | kNN  | Linear |
> |---------------|--------|----------------------------|-----------------------|------|--------|
> | DINO-vMF      | 100    | 8192                       | 684                   | 59.1 | 69.6   |
> | DINO-vMF (kd) | 100    | 8192                       | 1373                  | 59.8 | 70.4   |
> | DINO-vMF (kp) | 100    | 8192                       | 8192                  | 60.1 | 70.8   |

---

> > ### Comment · Reviewer_hqpg · 2023-11-22
> >
> > Thanks for the author's response. It addresses most of my concerns, and I will raise my score to 6.

---

### Author Response · Authors · 2023-11-19
**General response to all reviewers**

We thank all the reviewers for the valuable feedback and insightful suggestions to improve our paper. We have been silent until now since we have been running a few additional experiments to address the weaknesses raised by the reviewers. Based on the suggestions from all the reviewers, we have made the following updates to the paper:

- We have added a new section A.5 in the appendix with a few additional experiments to address the weaknesses raised by the reviewers.
- We show in section A.5.1 that the partial prototype collapse issue is not just limited to the DINO family and demonstrate that it also occurs in SwAV and for models pre-trained using EsViT based on different backbone architectures.
- We added section A.5.2 to provide an ablation experiment for the KoLeo-proto regularization strength to justify our choice in the experiments presented in the main paper.
- We demonstrate the effectiveness of the proposed KoLeo-proto regularization on an additional model-backbone combination in section A.5.3 using the DINO method and a Resnet50 backbone.
- We added section A.5.4 to provide a computational analysis of GPU memory use when using different numbers of prototypes. With a batch size of 100 per GPU, every additional 2048 prototypes require an additional 2.5 GB of GPU memory at fp16 precision. This sheds light on the importance of the effective utilization of the prototypes.
- We have added a section A.6 in the appendix to provide a visual explanation, complementing the textual explanations in the main paper. We use t-SNE plots to illustrate the impact of adding the KoLeo-proto regularization on the prototypes and the coarse/fine structure of clusters learned by the model. We also show examples of images from the different clusters for additional visual context.

We believe some of the explanations provided in the response to the reviewers can be valuable additions to the main paper. We plan to revise the main paper one more time with the following minor updates before the rebuttal deadline:
- We plan to incorporate a part of our explanation to reviewer hqpg, explaining the relation to [1] which proposes a fixed ETF geometric structure for the classification layer that is found to be beneficial for supervised long-tail classification.
- We will elaborate on the explanation for why the KoLeo-proto regularization is beneficial for long-tail classification in section 7
- As explained in the response to reviewer fvFf, we will add our motivation for focusing on the iBOT-vMF baseline in the experiments.

[1] Yang, Yibo, et al. "Inducing Neural Collapse in Imbalanced Learning: Do We Really Need a Learnable Classifier at the End of Deep Neural Network?." NeurIPS, 2022.

---

### Author Response · Authors · 2023-11-22
**Promised updates to the main paper are incorporated**

As promised in the earlier response, we have now incorporated into the main paper some of our explanations provided in the responses to the reviewers:
- In the related work section (section 5), we added a part of our response to reviewer hqpg, explaining the relation to [1] which proposes a fixed ETF geometric structure for the classification layer that is found to be beneficial for supervised long-tail classification.
- We elaborated on the explanation for why the KoLeo-proto regularization is beneficial for long-tail classification in the 2nd paragraph of section 7.
- We shortly motivate about our general focus on the DINO family of methods in the contributions part of the introduction section. For our primary contribution of defining and identifying partial prototype collapse, we investigate methods starting from SwAV to iBOT. Our secondary contribution is to study the downstream impact of preventing the partial prototype collapse through the proposed KoLeo-proto regularization. We demonstrate this on iBOT in the main paper and consider different pre-training datasets. We added a brief motivation for this choice in the beginning of section 6, based on our response to reviewer fvFf. In addition, we have showed the efficacy of our proposed KoLeo-proto regularization on DINO with a Resnet backbone during the rebuttal period (section A.5.3 in the appendix).

Again, we thank all the reviewers for the constructive feedback that helped in improving the paper further.

[1] Yang, Yibo, et al. "Inducing Neural Collapse in Imbalanced Learning: Do We Really Need a Learnable Classifier at the End of Deep Neural Network?." NeurIPS, 2022.

---

### Meta-Review · Area_Chair_8u2G · 2023-12-05

**Metareview:**

This paper presents a self-supervised learning algorithm. The key difference from the existing methods lies in the discovery and alleviation of the prototype collapse phenomenon.

After the rebuttal and AC-reviewer discussion stage, the final scores of this paper are 3/6/6/6. The AC asked for discussions, but only the negative reviewer (rating 3) responded and insisted on rejection, and all other reviewers did not respond. The AC looked into the case by reading the papers as well as reviews and the rebuttal. The following issues were found.
* This paper studied a clustering-based SSL framework which is closely related to contrastive learning. This framework is somewhat old and limits the potential of the paper's contribution to the community.
* The studied problem (prototype collapse) makes sense but it is not a key issue that harms the performance of SSL. This is evidenced by the limited accuracy gain in a few benchmarks.

In summary, this is a borderline case. The AC will finalize the decision after discussing with the SAC.


Updates by the PCs: SAC recommends rejection. Below is their meta review

The paper is concerned with self-supervised learning methods of the Dino type, which we can think of as discriminative clustering algorithms. The key observation is that all current methods learn fewer clusters than specified by the hyper-parameter K, because of partial collapse, in which some learned prototypes mu_k are equal to each other. They therefore propose to add a maxent regularizer (eqn 5) to the prototypes themselves, based on the "KoLeo" entropy estimator. This is related to, but slightly different from, the Koleo regularizer added to the data embeddings, as used in DinoV2. So the whole paper boils down to the proposal to add Eqn 5 regularizer on top of existing methods.

Fig 3a shows that the method works as intended, in the sense that the number M of learned unique prototypes increases with the upper bound K when using the proposed regularizer, but not with other methods. (See also the value of M in table 4 for inaturalist.) This suggests more efficient use of the representational capacity. Unfortunately, it seems that this improved representation has negligible benefit when it comes to using it for classification purposes, even for long-tailed, fine-grained problems. (There are some gains, but they are very small, and could well be due to things like hyper parameter tuning.)

As it stands, I believe the idea is too incremental, and the empirical results too underwhelming, for this paper to be accepted (echoing remarks of the AC and another reviewer). However, I encourage the authors to investigate other applications (beyond visual classification) for their proposed method, such as in scientific discovery, where estimating the 'correct' number of clusters may be more important.

**Justification For Why Not Higher Score:**

This paper does not study a pioneering topic and the experimental results show limited accuracy gains.

**Justification For Why Not Lower Score:**

Three reviewers (although not responding after the rebuttal) suggested acceptance.

---

### Decision · Program_Chairs · 2024-01-16

Reject